# Income and racial disparity in household publicly available electric vehicle infrastructure accessibility

Jiehong Lou ®[1] ✉, Xingchi Shen ®[2] ✉, Deb A. Niemeier[3] & Nathan Hultman ®[1]

Publicly available electric vehicle (EV) infrastructure is pivotal for the United States EV transition by 2030. Existing infrastructure lacks equitably distribution to low-income and underrepresented communities, impeding mass adoption. Our study, utilizing 2021 micro-level data from 121 million United States households, comprehensively examines income and racial disparities in EV infrastructure accessibility. Our analysis of national averages indicates that lower-income groups face less accessibility to public EV infrastructure in both urban and rural geographies. Black households experience less rural accessibility, but greater urban accessibility compared to White households conditioning on income. However, our localized analysis uncovers significant variations in accessibility gaps among counties, rural and urban settings, and dwelling types. While Black households experience greater urban accessibility nationally, a closer look at the county level reveals diminishing advantages. This study identifies areas with pronounced inequality and urgent needs for enhanced accessibility, emphasizing the necessity for tailored solutions by local governments to enhance equitable access to EV infrastructure.

Publicly available electric vehicle (EV) infrastructure (hereinafter referred to as "EV infrastructure"), plays an essential role in supporting the United States' (U.S.) 2030 transition to electric vehicles. The federal government has an ambitious goal: half of all new vehicles sold in 2030 should be zero-emissions vehicles and accompanied by a national network of 500,000 electric vehicle chargers[1]. A national network of 1.2 million chargers is necessary to accommodate an estimated 26 million EVs on the road by 2030[2]. According to several Integrated Assessment Models (IAMs), the number of EVs on the road in 2030 is predicted to be between 14 million to 60 million[3,4]. To achieve these targets and support additional EV market penetration, EV infrastructure must rapidly expand. The growth rate must reach an annual addition of 4 times and 11 times for each of the goals, respectively, compared to average annual additions between 2011 and 2022[5]. Growth this rapid can result in an inequitable distribution of both EVs and the underlying support infrastructure[2,6–8].

Public EV infrastructure offers unique contributions to EV deployment that private charging infrastructure does not. First, EV infrastructure promoted by the government serves as a non-monetary tool to incentivize EV demand[9]. Studies have shown that public charging infrastructure is strongly correlated with vehicle purchases[10–14]. Second, public EV infrastructure promotes future EV adoption by shaping preferences. When charging infrastructure is widely accessible, people tend to have a more positive perception of EVs and are more likely to choose them as their preferred mode of transportation[15]. Third, public EV infrastructure helps to reduce range anxiety for both daily use and longer-distance travel. Home charging improvements alone cannot facilitate long-distance trips, so the availability of fast public charging stations, in particular, is essential to boosting battery electric vehicle (BEV) sales[16–18]. Finally, public EV infrastructure helps to close the homeowner-renter gap for electric vehicle ownership[19].

[1]Center for Global Sustainability, School of Public Policy, University of Maryland, 7805 Regents Dr, College Park, MD 20742, USA. [2]School of the Environment, Yale University, 195 Prospect Street, New Haven, CT 06511, USA. [3]Dept. of Civil and Environmental Engineering, University of Maryland, 1188 Glenn L. Martin Hall, College Park, MD 20742, USA. ✉e-mail: jlou@umd.edu; xingchi.shen@yale.edu

Current EV adoption is constrained by limited charging infrastructure[18,20,21], the cost of infrastructure[10,22–24], site selection and planning[16,25–30], and grid capacity[31–34]. But research has also shown a disturbing trend of declining accessibility to EV infrastructure among low-income households, people of color in specific regions[2,29,35–39], as well as those residing in multi-family dwellings (MUDs)[40,41]. These trends have persisted at both the national and regional levels, with geographies scaled at the zip code or census block group (CBG)[41] and specific regions such as the state of California and New York City[38] where heterogeneity can obscure important variation. There is a significant gap in understanding of EV infrastructure at the individual, or microscale, level.

Recent legislative efforts, such as the Bipartisan Infrastructure Law, have flagged equity as an important consideration. Together, much of the legislation aims to facilitate the broader goal of mass adoption. Evidence suggests that lower-income communities will require consistent investments, equivalent to roughly 30% of chargers and charging investments through 2030, to ensure equitable infrastructure access[2]. Lacking a comprehensive (such as using granular household address level data with the national scale) understanding of race and income analysis disparities in access to public EV Infrastructure, policymakers cannot optimally direct EV infrastructure investment towards disadvantaged populations.

We use a detailed 2021 national dataset, consisting of micro-level data from 121 million households across 51 states to estimate the social disparity in public EV infrastructure accessibility. The concept of accessibility has its origins in the fields of transportation and urban planning, where it refers to the ease of reaching various destinations[42–44]. It has since evolved and extended its reach into the broader domain of social science, where the term "barrier" has been more frequently adopted. Our data, obtained from Data Axle, includes 97.5 % of households with valid addresses and provides residential location, estimated income, and racial/ethnic identifications. Equitable EV infrastructure encompasses various aspects, including affordability, distance, availability, reliability, safety, etc.[45] In this study, we specifically concentrate on addressing a crucial dimension of equity improvement, which is the proximity to charging stations. By emphasizing the distance to charging stations as a key factor, we aim to contribute to the literature on the equitable accessibility of EV infrastructure.

In this study, we employ four sets of research designs to analyze a distinct aspect of income and racial disparity in EV infrastructure accessibility. First, we examine the relationship between EV infrastructure accessibility and income for four different racial/ethnic households (Black, White, Hispanic, and Asian households) using the data of the national population. Our second research design maps the local spatial distribution of accessibility gaps at the nation-wide county level to identify geographical heterogeneity in racial disparity between Black and White households. We then use the least absolute shrinkage and selection operator (LASSO) regression to determine the relative importance of income and race identity in predicting EV infrastructure accessibility. Finally, we use our results to identify what matters for the income and racial disparity in EV infrastructure accessibility. Our findings reveal that the association between income and accessibility, as well as race and accessibility, is large and statistically significant. National analysis shows that lower-income groups (as defined in the Methods section for low-moderate income) are, on average, further from public EV infrastructure in both urban and rural geographies. We also find that, on average, Black households tend to have better accessibility to public EV infrastructure in urban areas, but worse accessibility in rural regions when compared to White households. However, our local analysis finds large spatial heterogeneity in accessibility gaps at the county level. Although Black households enjoy better urban accessibility on a national scale, a detailed examination at the county level reveals

that this advantage is diminished. In addition, income plays a more dominant role in the accessibility distribution when compared to racial identity in rural and single-family dwelling (SFD) types, while race plays a more influential role in accessibility distribution when compared to income in urban and multi-family dwelling types. Finally, our research suggests that local macro-level factors such as the disparity in proximity to the highway, the degree of local income inequality, and differences in dwelling types among different groups play significant roles in contributing to the accessibility gaps in EV infrastructure.

## Results

### Nation-wide social disparity in access to public EV infrastructure

Using locally weighted scatterplot smoothing (LOESS) regression and U.S. population data, we find a consistent (nonlinear) positive relationship between income and public EV infrastructure accessibility across four racial/ethnic households (Fig. 1a). In both urban and rural areas, as household income increases, access to the public charging station(s) also increases (the distance to the charging station decreases). Black households experience the least accessibility to public EV infrastructure in rural areas compared to White, Hispanic, and Asian households. In urban areas, Black households are the second least accessible group, followed by White households. Ordinary least squares (OLS) regressions are also conducted to compare accessibility among different racial/ethnic households while controlling for income and yielding consistent results (see Methods and Supplementary Table 6). On average, rural households have less access to EV infrastructure compared to their urban counterparts. These trends remain consistent even when up to 5 charging stations are considered.

Public EV infrastructure typically contains level 2 and Direct Current (DC) fast charging[46]. In this paper, we focus on infrastructure accessibility, specifically the proximity to charging stations. Potential bias may arise from consumers' preferences for DC fast charging stations over level 2 charging stations. To account for this, we undertake three additional robustness analyses to bolster our primary analysis, all of which are detailed in Supplementary Note 1 within the Supplementary Information (SI). One of our analyses introduces an index that accounts for users' preferences across various charging station types by assigning distinct weights to DC fast charging stations, another by examining the ratio of DC fast charging stations in our core analysis, and finally incorporates the analysis using the Inhabitant-to-Station Ratio through a two-tiered approach. The results consistently align with the overarching trends observed in our core assessment.

We also calculate the percentage increase in distance between the $N^{th}$ and $(N-1)^{th}$ nearest charging stations for households within each racial/ethnic population in the top and bottom 5% of income. Our results, presented in Figure 1b, indicate that in urban areas, households in the lowest income group experienced a greater increase in distance to additional charging stations compared to those in the highest income group (top 5% of income), regardless of their racial/ethnic background. Interestingly, we find that this trend is consistent for Asian and Hispanic households in rural areas as well. In summary, the results highlight the disparities in accessing additional charging stations based on income and race or ethnicity.

### Income and racial disparity in county accessibility gaps in public EV infrastructure

To evaluate equity at the nationwide county level, we quantify two types of accessibility gaps: one related to income disparities and the other centered around racial disparities as our core dimensions of accessibility gaps. We have chosen income and race because they represent two structurally rooted problems. Income is a fundamental metric to measure people's economic ability and social class. Racial inequality and discrimination is another deeply rooted, systemic

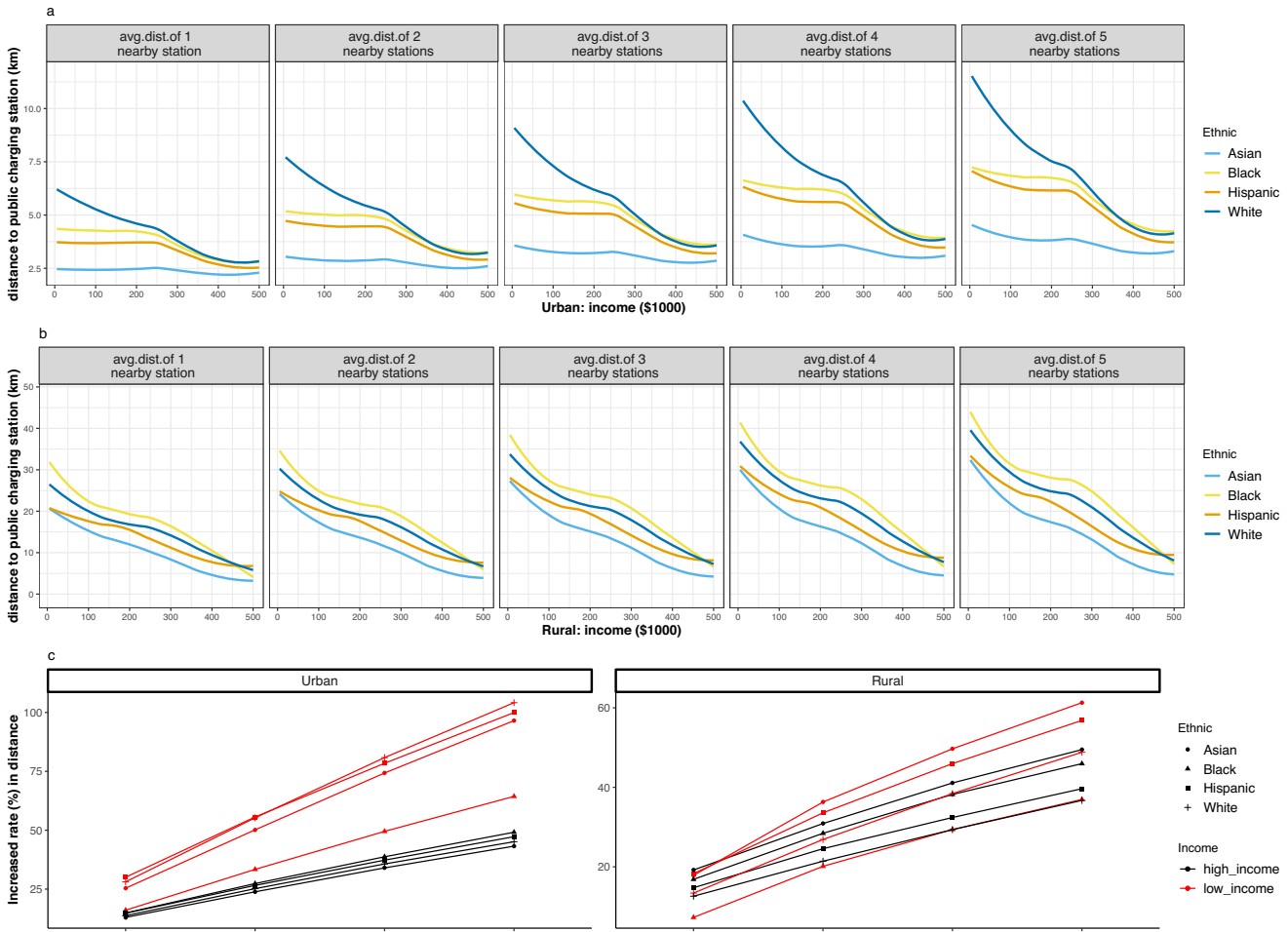

**Fig. 1 | Nation-wide electric vehicle (EV) infrastructure accessibility gap.**
**a** Locally weighted scatterplot smoothing (LOESS) regression curves displaying the relationship between income and the average distance to the N-nearest EV infrastructure among different racial/ethnic households in urban areas. **b** LOESS regression curves among different racial/ethnic households by average distance to N-nearest EV infrastructure in rural areas. **c** Increased rates (%) between Nth and (N-1)th nearest EV infrastructure accessibility among different racial/ethnic households and income groups in urban and rural areas. EV public infrastructure accessibility showing the LOESS regression curves by $100,000 interval of the entire sample ($n = 118,682,791$). The LOESS regression curve, represented by the solid line in the center, shows the predictive value, while the shadow around it represents the 95% confidence interval.

problem in the U.S. A vast literature focuses on these two factors[19,39,47–50]. More practically, income and race are two key indicators we can observe at the household level nationwide in our dataset.

To measure the first core gap, we use the difference in distance to the EV infrastructure for low-moderate income (LMI) and non-LMI households (Fig. 2a and b). To measure the second core gap, we use the difference in distance to the EV infrastructure for the Black and White households (Fig. 2c and d). In this analysis, we focus on the racial accessibility gaps between Black and White households instead of all types of racial/ethnic households. Our interest in the spatial differences between Black and White households stems from the persisting disparities and inequities faced in accessing social resources, including transportation infrastructure, health, education, and more[51–53]. Here, positive numbers indicate that LMI and Black households experienced larger accessibility gaps (or barriers) compared to non-LMI and White households, respectively. Negative numbers suggest that LMI and Black household experience a lower accessibility gap compared to White households.

As might be expected, we see mixed accessibility gaps between LMI and non-LMI households at the county level in both rural and urban areas. We observe that more counties have LMI households facing accessibility gaps in rural areas (Fig. 2a) than in urban areas (Fig. 2b). 53% of the county had LMI residents experiencing accessibility gaps in rural areas compared to 35% of counties in urban areas. In rural areas, LMI households in counties with accessibility gaps have to travel an average of 2.92 km more to reach EV infrastructure compared to non-LMI households. In urban areas (Fig. 2b), LMI residents in counties with accessibility gaps have to travel more on average to reach EV infrastructure (1.73 km), compared to those LMI residents in counties without accessibility gaps (travel less by 1.55 km). This is consistent with the literature showing that low-income households are more likely to live in the center of the city in the urban areas[54–56].

The accessibility gaps between Black and White households at the county level are mixed. In about half of the rural areas in counties (49%), Black households have to travel an average of 2.30 km more to reach EV infrastructure than White households (Fig. 2c). While in urban areas, about 34% of counties have Black households experiencing accessibility gaps (Fig. 2d). Although the inequity between different racial households is less pronounced, Black households have to travel further on average when experiencing accessibility gaps relative to their counterparts who do not experience such gaps. Specifically, Black households have to travel 1.20 km more to EV infrastructure, when experiencing accessibility gaps. In contrast, when White households have accessibility gaps, they travel an average of 1.10 km more to the EV infrastructure.

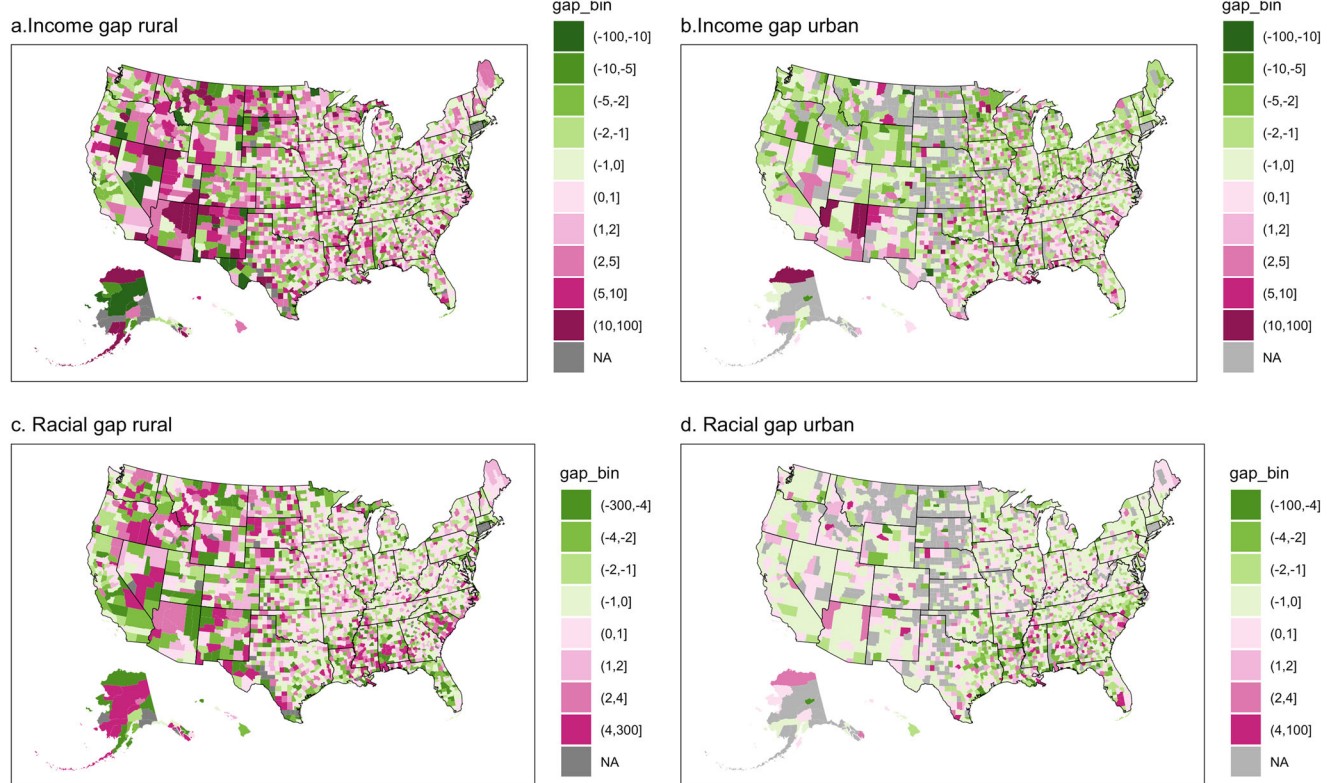

**Fig. 2 | Equity assessment through accessibility gaps at county level. a, b**: Accessibility gaps between low-moderate income (LMI) and non-LMI households by rural and urban, respectively; **c, d** Accessibility gaps between Black and White households by rural and urban, respectively. The magenta color scheme shows that LMI and Black households experienced larger accessibility gaps (or barriers) compared to non-LMI and White households, respectively. Vice versa, the green color suggests the opposite. Gray color means no data available or beyond the defined minimum and maximum thresholds of −300 and 300 km, respectively.

Public resources, specifically the public EV infrastructure, should be equitably accessible to all demographic groups, irrespective of their income, wealth, or racial backgrounds. Our county-level analysis reveals mixed disparities in local access that diverge significantly from previous studies using averages and larger geographies[2,57]. These disparities underscore the necessity of developing policies tailored to local conditions. Our findings suggest that regions marked in red require more substantial policy support. This stems from the recognition that low-income and Black populations are vulnerable groups. For example, rural areas, depicted in red, necessitate increased support.

Finally, we recognize the existence of various dimensions related to accessibility gaps, encompassing factors like vulnerability, social and economic disadvantage, and educational attainment, among others. In our pursuit of an exhaustive analysis of the accessibility gaps from income and racial dimensions, we conducted an additional investigation by incorporating the Electric Vehicle Charging Justice40 Map Tool, jointly proposed by the Department of Energy and Department of Transportation to explore other equity dimensions. This tool, designed to identify Disadvantaged Communities (DAC) at the census tract level, offers an alternative perspective on accessibility gaps (refer to Supplementary Note 2) between DAC and non-DAC. Our examination revealed strong correlations between our racial accessibility gap and the DAC accessibility gap, as well as between our income accessibility gap and the DAC accessibility gap, both at the county and state levels. It is crucial to underscore that our analyses of income and racial disparities are firmly grounded in individual household data, providing a finer-grained understanding compared to analyses conducted using demographic data at the broader census tract level. This heightened level of detail enhances our confidence in the robustness of our findings, particularly when comparing them with the DAC accessibility gap.

### Income and race play different dominant roles in the prediction

Using the LASSO machine learning approach, we explored whether income or race was a better predictor of access to public EV infrastructure accessibility. We find compelling evidence that both race and income significantly influence the accessibility of public EV infrastructure. However, the relative importance of these two factors exhibits geographical heterogeneity, especially in the urban and rural, MUD and SFD settings. One intriguing finding from this analysis is the pronounced association between race and the presence of EV infrastructure in MUD compared to SFD, both in rural and urban settings. In urban areas specifically, race emerges as a more influential factor.

In rural areas, income emerges as a better predictor for accessibility than race in most states. For rural SFD residents, income is a better predictor of EV infrastructure accessibility in 48 out of 50 states (Fig. 3a). The dark green color in the figure represents a negative correlation between distance to the EV infrastructure and income, indicating that households with higher income have better accessibility to EV infrastructure, as they have to travel a shorter distance to reach it. Hawaii and Alaska have different outcomes due to their special features, and they can be considered outliers. However, for rural MUD residents, we start to observe that race plays a more dominant role compared to income in a few states. In 7 out of 50 states (Fig. 3c), race is associated with a positive correlation, and has a higher predictive power compared to income, indicating that when a MUD resident belongs to the Black minority group, the distance to the EV infrastructure increases.

In urban areas, our findings suggest that race increasingly serves as a more important predictor of EV infrastructure accessibility across a larger number of states. For urban MUD residents, 12 out of 51 states demonstrate a positive and stronger association

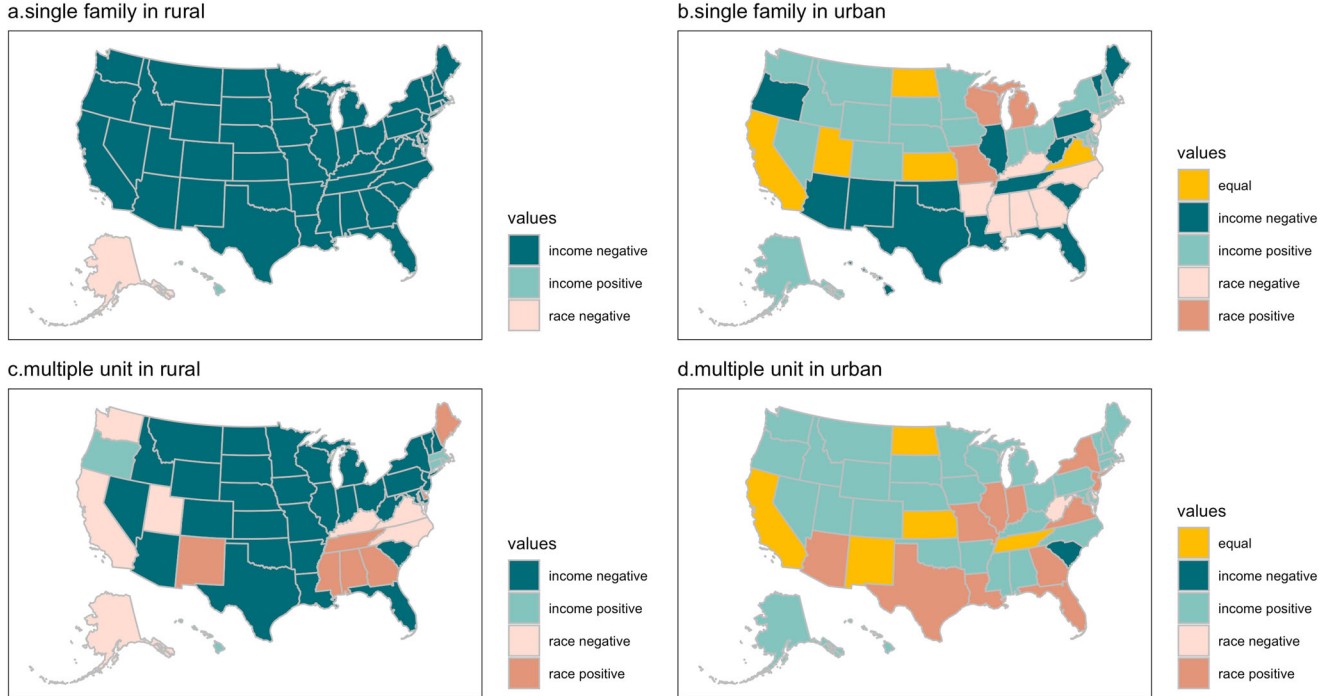

**Fig. 3 | State level least absolute shrinkage and selection operator (LASSO) coefficients across locations and house types. a** Rural single-family dwelling (SFD) residents. **b** Urban SFD residents. **c** Rural multi-unit dwelling (MUD) residents. **d** Urban MUD residents. Note that the District of Columbia (DC) is not shown in panels (a) and (c) since all DC residents are classified as urban. The coefficients are color-coded to indicate their direction (positive or negative). When the coefficient estimate of income or race (being Black households) is positive, it means that a higher value of that factor is associated with a longer distance to the nearest electric vehicle (EV) infrastructure, while a negative coefficient estimate suggests the opposite. Equal means coefficient estimates of income and race approximate towards zero at the same time with the increased alphas.

between race (being Black households) and distance to EV infrastructure (Fig. 3d). This could be due to certain reasons in these 12 states: Black households are more likely to reside in urban MUDs, yet their access to public EV infrastructure is considerably limited when compared to White households. This finding aligns with existing literature in that underrepresented groups often find themselves concentrated in impoverished urban areas[58–60]. Therefore, it is crucial to acknowledge potential interaction effects between the spatial location of most public EV infrastructure and the residential arrangement of both Black and White neighborhoods, with a greater proportion of low-income Black households in urban settings.

For a more comprehensive understanding of EV infrastructure accessibility, we extended our LASSO analysis to include four additional outcomes (average distance to 2 nearest stations (DP2)- average distance to 5 nearest stations (DP5)). These outcomes include measures such as the average distance to two nearby stations, and up to five nearby stations. Our findings consistently highlight, in line with the previous discussion, the significance of factors such as income, race, housing type, and location. For a detailed presentation of these findings, please refer to Supplementary Fig. 9-12. The results obtained through our machine learning approach reinforce and are consistent with the conclusions drawn from our previous correlation analysis, which emphasized the importance of income and race as key predictors of EV infrastructure accessibility but also shed light on the access disparities that existed between SFD and MUD. Aligning with the literature that lower-income households, or underrepresented households are more likely to reside in MUD, our results further confirm that in many states, especially in urban areas, Black households experience accessibility issues to public EV charging stations.

## Potential contributors to the income and racial disparities in public EV infrastructure

Our previous analysis revealed that accessibility gaps vary widely across different counties. To shed light on the causes of these heterogeneous accessibility gaps, we delve into the potential factors that have a strong correlation with income and racial disparities in EV infrastructure accessibility, employing a regression-based framework in this section. Our approach involves examining the strong correlation between the county-level accessibility gap and a set of predictor variables at the county level. Factors with strong correlations help to explain the accessibility gap. We conduct two separate OLS regressions, with one focusing on the accessibility gap in income and the other on the accessibility gap in race. Our variables derive from a systematic literature review (see Supplementary Note 4) and our methodology can be seen in full detail in the Methods section.

To visually represent the results, we present the estimated coefficients of interest in Fig. 4, and the corresponding table output in Supplementary Table 7. Panels a and b depict the estimated results for the income-based EV infrastructure gap. Our analysis shows that the highway gap is a significant contributor to the accessibility gap, followed by the income gap. The positive association between the highway gap and accessibility gap between LMI residents and non-LMI residents indicates that if LMI lives farther from the highway than non-LMI residents, accessibility to public EV infrastructure will decrease. The negative association between the income gap and accessibility gap indicates that if the income gap between LMI and non-LMI residents increases, the accessibility gap will widen. Moreover, we observe that both the MUD gap and SFD gap contribute to the accessibility gap. However, we note the opposite signs of MUD gaps between rural and urban settings, which can be attributed to the higher number of charging stations located around MUDs in urban areas.

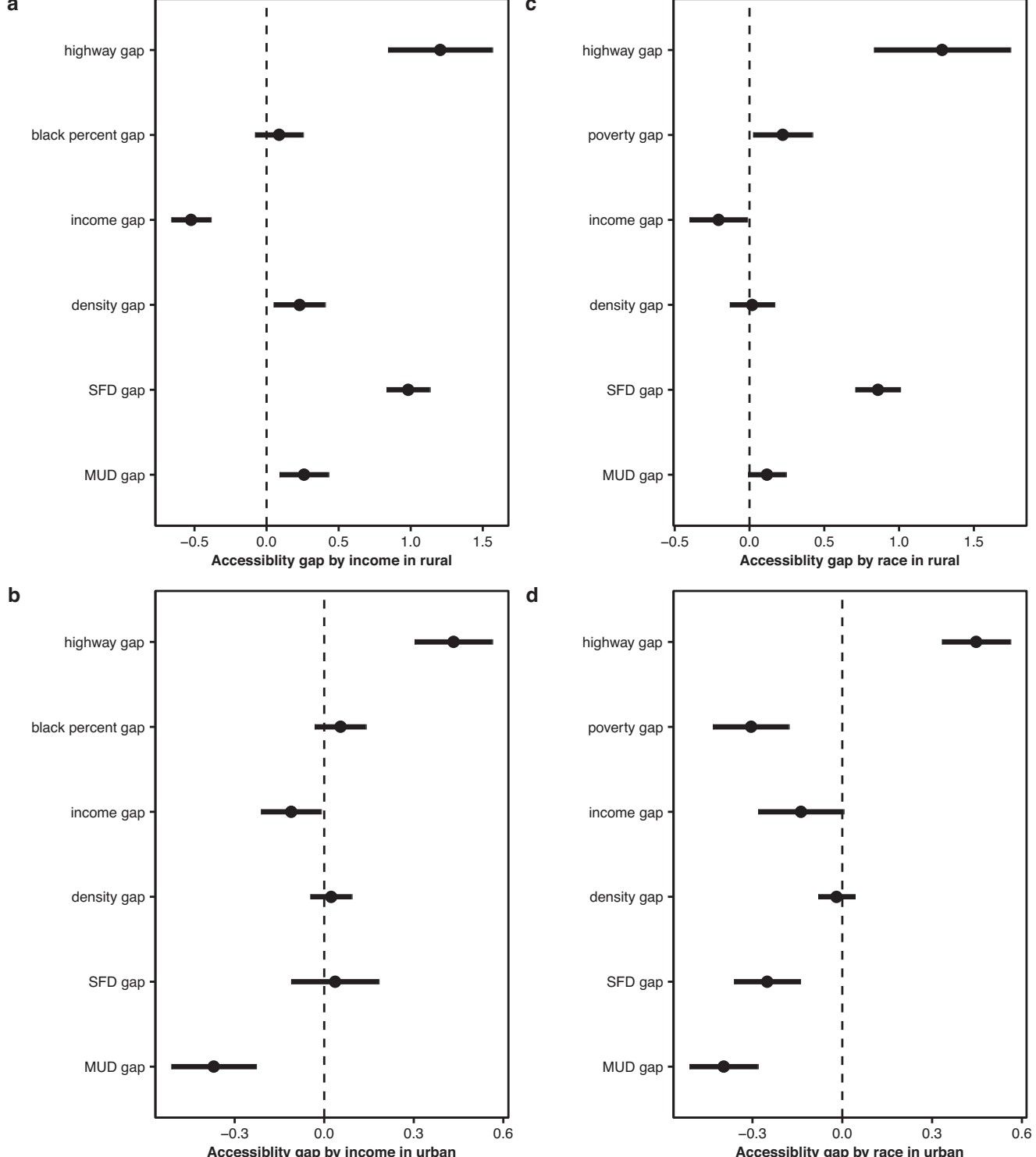

**Fig. 4 | Point estimates of each factor influencing social gaps in electric vehicle (EV) infrastructure, with error bars.** Infrastructure gap by income with coefficients of influencing factor ($n = 2362$ in rural areas in panel **a**, $n = 2348$ in urban areas in panel **b**). Infrastructure gap by race with coefficients of influencing factors ($n = 1937$ in rural areas in panel **c**, $n = 2358$ in urban areas in panel **d**). All the variables, except for the outcome variable, are normalized before running the regression. The black dots indicate the changes in EV infrastructure gap obtained from running separate regression analyses. The black horizontal bars represent the 90% confidence intervals of the estimations. Therefore, data are presented as coefficient values+/−1.65*standard error. "MUD" means multi-unit dwelling and "SFD" means single-family dwelling.

To provide a visual representation of our regression-based framework for the accessibility gap between Black and White households, we present Figs. 4c and 4d. Our results indicate that the highway gap is the primary contributor to the accessibility gap, followed by the income gap. Moreover, we observe that the poverty gap between Black and White households (i.e., the ratio of the poverty population in each group) also plays a significant role in contributing to the accessibility gap, but in opposite directions in rural and urban areas. Specifically,

larger poverty gaps increase the accessibility gap in rural settings, while decreasing it in urban settings. Once again, we observe opposite signs for MUD gaps and SFD gaps between the rural and urban settings. This finding is likely due to the distribution of public charging stations around MUD areas in urban areas, where there is also a larger poverty population.

Overall, our results highlight the significant role of the highway gap, income gap, and poverty gap in determining the accessibility gap between LMI and non-LMI residents, and Black and White residents. They also demonstrate the importance of considering both MUD and SFD gaps in addressing the accessibility gap in both rural and urban settings.

## Discussion

Despite the fact that 80% of EV owners in the United States charge their vehicles at home, the presence of public EV infrastructure can help alleviate challenges faced by certain population groups, particularly urban and MUD residents. For example, residents of apartments may not have access to home charging options and may rely solely on public EV infrastructure for vehicle charging[29]. Additionally, research shows that urban groups tend to have lower rates of home charging[29]. While home charging infrastructure may be sufficient for the early market diffusion of EVs, there is a growing need for more public EV infrastructure to support future market growth[20] and to ensure the availability of EV technology across race/ethnicity and population groups.

Low-income and underrepresented households (non-White racial/ethnic households) face numerous challenges in accessing private EV chargers. These households constitute a smaller proportion of key demographic subgroups compared to their white counterparts in the general population. Low-income and underrepresented people are more likely to live in MUD housing units and less likely to install a private EV charger compared to high-income groups and White groups, who are more likely to live in SFD units. Low-income and underrepresented groups are more likely to rent homes. Empirical data supports these disparities. According to our analysis, 68% of Black populations live in SFD units, compared to 81% of White populations. Additionally, only 71% of low-income populations own their homes, compared to 90% of high-income populations. When looking at the percentage of each group that rents their homes, we see an even more significant difference: 63% of Black populations are renters, compared to only 30% of White populations. These disparities highlight the need for policy and infrastructure improvements to ensure that low-income and underrepresented households can access the same amenities as their higher-income and White counterparts.

While previous studies have highlighted the lack of private charging stations as a major concern[16], our research demonstrates that there are additional challenges for these households in accessing public charging infrastructure. This is particularly true for those who reside in MUDs, where charging stations are less likely to be available. Given the growing demand for EVs and projections that this trend will continue, it's essential that we find solutions to these challenges. One promising development is that 42% of the increase in EV infrastructure by 2030 will be allocated to MUDs[2], which will help to address the current lack of charging stations in these areas. However, it is important to note that additional measures must be taken to ensure that low-income and underrepresented households have equitable access to public and private EV infrastructure. This could involve public investment in charging infrastructure in areas with high concentrations of these households, as well as policies mandating the inclusion of charging infrastructure in new residential and commercial developments.

The current federal and state policies related to EV public infrastructure emphasize increasing the availability of charging stations and providing incentives for their installation. Some examples of current federal policies are the National Electric Vehicle Infrastructure (NEVI) Formula Program which aims to fund the installation of alternative fuel corridors, including EV charging stations, along major highways in the United States. The General Services Administration (GSA) has established a program to install EV charging stations at federal buildings and properties. At the state level, many states offer rebates or tax credits for the purchase of EVs and for the installation of EV charging stations and mandate that new buildings or parking facilities be constructed with EV charging infrastructure. However, the gap present in these policies is that most focused on broader installation, missing the equity issues. A crucial aspect of this transition is the strategic development of EV charging stations in economically disadvantaged regions, ensuring that the benefits of electric transportation are accessible to all segments of society. We conducted an extensive analysis of EV charging regulations and incentives at the federal and state government levels (see Supplementary Note 3 for the comprehensive database). We found that there are only 26 existing regulations aiming to promote equitable access to electric vehicle charging infrastructure implemented in 16 states. In contrast, incentives are more common in states across the US. This indicated a clear gap in regulations from the state government which suggests that the subnational government should implement tailored solutions. Given that public EV infrastructure is of significant national interest, it is crucial for this market to be well-regulated by public authorities to ensure access for all.

On the policy implantation side, our county-level analysis provides additional insights into the current accessibility gap with respect to income and locality, which differs slightly from the state-level findings presented in Fig. 5. However, it is important to note that the state-level results can be misleading in certain aspects. Firstly, while state analyses emphasize the disparities between low- and non-low-income households in rural areas (Fig. 5a), they fail to capture the nuanced variations within each state. Secondly, the state-level analysis obscures the significant accessibility disparities between Black and White households in these states, as the gap becomes less visible due to aggregation (Fig. 5d). Our findings reveal two important observations. On the one hand, the persistent accessibility gaps in rural areas observed at the state level are less pronounced at the county level. On the other hand, the persistent accessibility gaps in urban areas observed at the state level are more pronounced at the county level. Consequently, our results carry crucial policy implications for state governments, as they are typically better positioned to implement policies regarding public EV infrastructure[61]. Relying on misleading results might result in missed opportunities for states to contribute to the battle for equitable access.

Our results clearly demonstrate that there is a disparity in the deployment of EV infrastructure in disadvantaged communities, which is not adequately addressed by existing federal and state policies. We also find that EV infrastructure distribution differs significantly between rural and urban settings and between different dwelling types. Tailored solutions are needed to improve EV access and adoption in both settings, which will contribute to the 2030 goal of building an equitable network of EV chargers accessible to all households. Our findings also highlight the importance of addressing issues of equity and access in the context of the Equitable and Transformative Investment Framework[35], particularly given the significant investment needs in EV infrastructure. To ensure equitable access to EV infrastructure for all households, regardless of income or racial/ ethnic background, targeted policies and investments are needed.

It is important to note that there are a few caveats to our results. First, we use the travel distance between the residential addresses to the charging stations, as the dimension of the accessibility. This is only one aspect of accessibility that could be explored, and our results do not necessarily represent a full picture of the accessibility dimensions. For example, significant congestion and workplace location can result

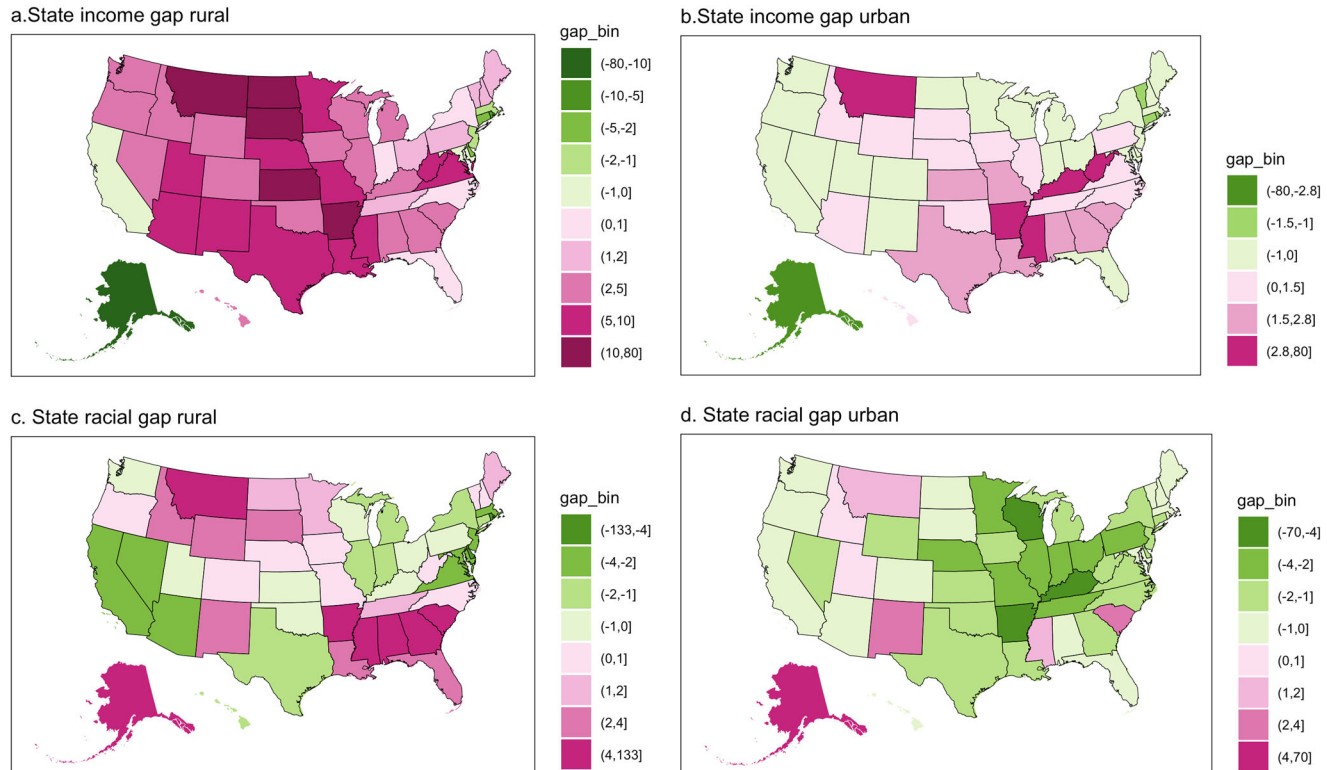

**Fig. 5 | Equity assessment through accessibility gaps at state level.**
**a**, **b** Accessibility gaps between low-moderate income (LMI) and non-LMI households by their location at the state level; **c**, **d** Accessibility gaps between Black and White households by their location at the state level. The magenta color scheme shows that LMI and Black households experienced larger accessibility gaps (or barriers) compared to non-LMI and White households, respectively. Vice versa, the green color suggests the opposite. Gray color means no data available or beyond the defined minimum and maximum thresholds of −133 and 133 km, respectively.

in accessibility declines. Second, while our paper primarily delves into two distinct accessibility gaps, one associated with income disparities and the other centered around racial disparities, we are acutely aware of the potential correlation between race and income. Our county-level regression analysis in the section on potential contributors to the income and racial disparities in public EV infrastructure illustrates that the income gap significantly influences the accessibility gap in the context of racial analysis.

## Methods
### Data
Household-level demographics data (Data Axle)[62]. We obtained household-level demographics data from Data Axle. To prevent double counting issues, we include only households indicating their current addresses as their primary addresses. This resulted in 120,962,661 observations, accounting for approximately 97.5% of households in the United States. We then focused our study on Black and White households, resulting in 94,878,711 observations, which accounts for 77% of the households in the United States. These data include information on the household's projected income, racial/ethnic identifications, and type of dwelling. A detailed summary of the data can be found in Supplementary Table 2. This level of granularity empowers us to distinguish and categorize households by income and race/ethnicity. In cases where households comprise multiple races/ethnicities, we have selected the head of the household's race/ethnicity for classification. Additionally, to ensure that our manuscript maintains unbiased language when referring to race/ethnicity, we have provided a detailed description in Supplementary Note 5. Notably, Data Axle has undergone a meticulous reconstruction process, integrating the most recent income statistics from the U.S. Census Bureau, self-reported survey data, real-time shifts in

earned income dynamics, and updates within the Consumer Database itself. The modeling framework is grounded in a robust foundation of statistical analysis, culminating in final estimates that are adjusted at both the county and block group levels. These adjustments are made to ensure a more accurate representation of income distributions, aligning closely with the most current Census data. To verify the reliability of Data Axle data, we conducted income and race/ethnicity correlation analysis with the U.S. Census American Community Survey data at the zip code level and found that they highly and positively correlated in Supplementary Table 4. The study received exemption from the institutional review board (IRB) at the University of Maryland with the number (#2176750-1) under the Exemption category #45CFR46.104(d)(4)(ii).

EV public infrastructure data (Public charging stations). In this paper, we focus on publicly available electric vehicle infrastructure. We will use the term "private EV charger" in this paper to differentiate it from public EV infrastructure. Private chargers are defined as charging stations that are located in homes, workplaces, or other private settings and might have access or technology limitations. We utilize the publicly available Electric Vehicle Charging Station Locations dataset from the Alternative Fuel Data Centre at the U.S. Department of Energy[63] as our primary source of EV infrastructure data. We extracted all the charging stations located in the United States from the dataset, resulting in 56,940 stations with 128,964 Electric vehicle supply equipment (EVSE) ports, 88% of which are powered by electricity.

Other demographic and economic data. (1) Rural and urban definition. We utilized the 2010 Census Urban and Rural Classification and Urban Area Criteria from the U.S. Census Bureau[64]. Our study includes both the urbanized areas (UAs, 50,000 or more people) and Urban Clusters (UCs, between 2500 and 50,000) as the urban areas. This definition encompasses residents from cities, suburbs, and towns,

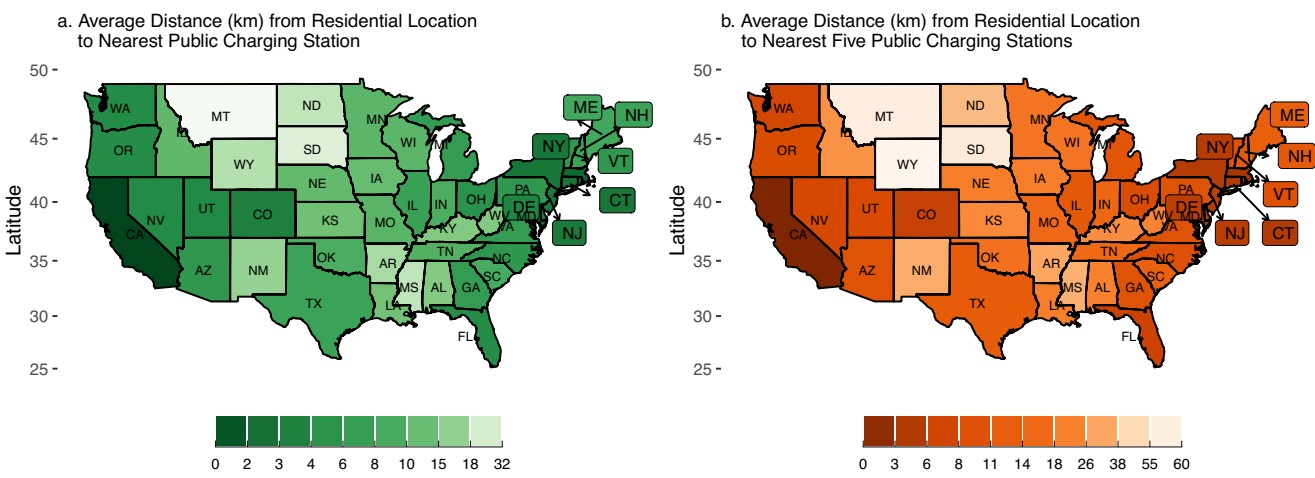

**Fig. 6 | Electric vehicle (EV) infrastructure accessibility for U.S. households. a** Average distance (km) of all the households to the nearest EV public charging station by state. **b** Average distance of all the households to the nearest 5 public charging stations.

aligning seamlessly with the town, suburban, and city categories suggested by the Department of Education, National Center for Education Statistics (NCES) definition (refer to Supplementary Fig. 21b)[65]. We compiled the 2010 Urban Area list file from Census based on the urban area census code (UACE) and converted all the UACE to the zip code level using the 2010 Urban Area to ZIP Code Tabulation Area (ZCTA) Relationship File. (2) LMI. Low- and moderate-income thresholds for each county are obtained from the Tool for Determining LMI Households provided by the U.S. Department of the Treasury (U.S. Department of Treasury, 2021). To streamline our analysis, we have adopted the moderate-income threshold for a typical three-member household for each county as recommended by the Tool. (3) Distance to the nearest freeway or highway. We calculate the shortest distance from the household residential location to the nearest freeway or highway using the Python Geopandas package. 4) MUD housing unit rate: calculated as the total MUD units divided by the total housing units in each county. In addition, we draw the following socioeconomic and political variables from the county level, such as population density, medium household income, medium house age, etc., from the U.S. Census County level from the 2021 ACS five-year estimates. For a detailed summary, please see Supplementary Table 5.

Summary Statistics: EV infrastructure accessibility overview. To examine spatial heterogeneity, we combine public EV infrastructure data from the Alternative Fuel Station Locators, which locates public EV charging stations, with our existing nationwide household dataset. We compute the shortest distance between households and up to five of the nearest public charging stations. Our analysis indicates significant variations in public EV infrastructure accessibility across states (Fig. 6). The distance to the nearest public EV charging station ranges from as little as 0.97 km in DC to 31.17 km in Montana (Fig. 6a). We also observe an increasing trend in distance from the nearest charging station to the distance of the up to five nearest charging stations. As might be expected accessibility to public charging infrastructure is highest on the west and east coasts. Our Supplementary Table 1, which calculates up to 10 closest charging stations, confirms these observations. We find that these trends remain consistent even with up to 10 charging stations considered (Supplementary Fig. 1).

**Empirical strategies**

Research objectives and contributions. In this paper, we outline two primary research objectives. First, we are interested in quantifying the differential accessibility to public EV charging infrastructure by income and race, two critical factors that arise in the literature. Second, we aim to provide an analysis of the geographic heterogeneity of accessibility

gaps across different counties and states. Finally, we seek to understand the origins of disparities and explore potential strategies for narrowing the gap. To address our first research objective, we have formulated two sub-tasks. First, we inquire whether the current EV infrastructure network offers equitable access to all residents. Second, we investigate the respective roles of income and racial disparities in shaping the accessibility of EV infrastructure. It's noteworthy that, for our research questions, we refrain from positing specific hypotheses. We adopt an exploratory stance, presenting results as they unfold, consistent with the emerging research on EV adoption in general.

The existing body of literature consistently highlights a troubling trend: diminishing accessibility to EV infrastructure among low-income households, individuals from underrepresented racial/ethnic backgrounds[2,29,35–39], as well as those residing in multi-family dwellings[29,41]. We also know that these trends in certain locations (e.g., California and New York City) and for certain higher-level geographies (e.g., some zip codes). There is, however, a significant gap in first, understanding trends using a highly resolved unit of analysis (i.e., individual or microscale level) and second, understanding of how these patterns vary nationally. Our paper directly contributes to the existing body of knowledge by presenting a national analysis enriched with individual household-level data.

We implemented four research designs to investigate the income and racial disparity in the accessibility of EV public infrastructure. First, we utilized LOESS to depict the nationwide relationship between EV infrastructure and income by four racial/ethnic households (Black, White, Hispanic, and Asian households). Second, we mapped out the heterogeneity of accessibility gaps between White and Black households as well as between LMI and non-LMI households at the nationwide state and county levels. Third, we used a machine learning approach to identify the relative importance of income and race in predicting the EV infrastructure accessibility. Finally, we aggregated our data at the county level and assessed the possible factors driving the observed income and racial gap in EV accessibility, respectively. This allowed us to identify the primary drivers of these gaps and provide policy implications.

Estimation of the nonlinear association between accessibility and income by race/ethnicity. We identify the overall social disparity in EV infrastructure accessibility using a pooled nationwide dataset covering 121 million households across 51 states. To account for the nonlinearity and identify the correlation between income and EV public infrastructure accessibility among different racial/ethnic households, we utilized LOESS regression. As a non-parametric strategy, LOESS is utilized to present a graphical representation of the relationship between

the dependent variable, EV infrastructure accessibility, and our independent variables of interest, income, and racial/ethnic identity[66]. In contrast to traditional statistical regression models, LOESS can uncover the relationships between the dependent and independent variables by fitting a nonlinear smooth curve, thus illustrating the central dependency of the dependent variable on the distribution of the independent variable[66]. The LOESS model is data-driven, and the fitting algorithm requires a smooth curve to pass through the densest area of the data. We utilized the R ggplot2 package's built-in function to create the desired LOESS scatter plot. We first analyzed the LOESS regression curves in the full sample, as shown in Fig. 1a. To enhance the robustness of our results, we performed the LOESS regression on a more refined sample by removing the households in the top and bottom 5% of income to mitigate the influence of extreme values in Supplementary Fig. 2. The refined sample exhibited a similar pattern to our main analysis. Additionally, we performed a regression analysis to investigate the correlation between accessibility and race/ethnicity conditioning on income. The findings, as presented in Supplementary Table 6, provide additional support to the LOESS regression results.

Determining the relative importance of income and race. We utilized a machine learning approach called LASSO, to examine which factors, income or race, more consistently predict the accessibility of EV infrastructure. LASSO is a data-driven method that shrinks some coefficients and reduces the value of others towards zero due to its penalty, effectively removing coefficient estimates that have little contribution to the model. We allowed a set of different values of L1 norm to be used in the analysis, with the initial LASSO models containing many predictors with high magnitudes of coefficient estimates. As the L1 norms increased, the coefficient estimates eventually approached zero, as illustrated in Supplementary Fig. 13 using data from the state of Georgia. To ensure that the models fully utilized the rich dataset, we included all household-level variables such as distance to the highway (dist), number of children (children), duration of residence (residence), and rural or urban status. From this process, we were able to identify which variables have consistent and predictive power. To determine which variable - income or race - had better predictive power, we residualized the county fixed effects, distance to the highway, number of children, and duration of residence, before performing the LASSO model on the remaining two variables. This allowed us to clearly identify the variable with the stronger predictive power for the accessibility of EV infrastructure. Supplementary Figs. 14 present the resulting plots of the LASSO technique using Georgia as an example, and Supplementary Figs. 15-18 are for each state, respectively, while residualizing county fixed effects and the other variables mentioned above. Finally, to present our main results in a more readable way, we created maps showing the relationship between income and race in the Results section. We compared income and race in each state to determine which variable had a stronger predictive power for EV infrastructure accessibility. This determination was based on the variable selection process, where we assessed which variable persisted as a significant predictor for a longer duration. We then determined whether this variable's coefficient was positive or negative and colored the state accordingly using a sign indicator on the map. For example, if income had a stronger predictive power and the coefficient was negative, we colored the state dark teal in Fig. 3 and labeled it as "income negative". By visually presenting the dominant predictor variable for each state, our maps highlight the importance of considering both income and race in efforts to promote equitable access to EV infrastructure. Furthermore, these maps provide a clear and concise summary of our findings that can be easily interpreted by a wide audience.

Regression-based mechanism analysis. We adopted OLS regressions at the county level to analyze the potential factors contributing to the racial and income gaps in EV infrastructure accessibility. Our dependent variables were the racial and income gaps at the county level. We selected independent variables from systematic literature

and transformed them into gap formats (i.e., the gap between LMI and non-LMI and the gap between Black and White residents) to incorporate them into our regression models. Previous studies suggest that the distance to the highway[38,41], multi-unit dwelling rates[41], and median household income[38,41] affect the installations of EV infrastructure. We computed the distance from residential addresses to the highway, MUD and SFD ratios, population density, and average income by income and race (Supplementary Figs. 19 and 20). We then calculated the gaps of these variables between income-based and racial households and normalized all the explaining variables before running our regression models. This normalization approach facilitated the comparison of the coefficients of different variables. Our approach is described by the following regression model:

$$Y_{ij} = \gamma + \delta V_{ij} + \varphi_j + \varepsilon_{it} \tag{1}$$

where $Y_i$ is the EV accessibility gap in county $i$ state $j$. $V_{ij}$ is a vector of all the gap-related variables, including the gap in distance from residential addresses to the highway, in MUD ratios, in SFD ratios, in average income, in density, in poverty ratio (for racial gap only), and in the Black ratio (for income gap only). $\varphi_j$ controls for state fixed effects capturing all the time-invariant state-specific characteristics. $\varepsilon_{ij}$ is an idiosyncratic error term.

### Reporting summary
Further information on research design is available in the Nature Portfolio Reporting Summary linked to this article.

## Data availability
This paper employs household-level demographic data sourced from Data Axle under a confidentiality agreement that prohibits the sharing of raw data. All the other data used for this analysis is available from publicly available sources cited or from the authors upon request. Source data for figures in the main text are provided in this paper. All the sharable data are available on GitHub from:https://github.com/Jiehonglou/Income-and-Racial-Disparity-in-Household-Publicly-Available-EV-Infrastructure-Accessibility Source data are provided with this paper.

## Code availability
All data and models are processed in R, Python, and Stata 17. The figures are produced in Python, R, and QGIS. All the relevant codes are available on GitHub from: https://github.com/Jiehonglou/Income-and-Racial-Disparity-in-Household-Publicly-Available-EV-Infrastructure-Accessibility.

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

## Acknowledgements
J.L. and N.H. acknowledge support from Bloomberg Philanthropies (307334-00001). We extend our gratitude to the Research Assistants Aayushi Gupta and Shawn Edelstein for their valuable research support in collecting policy data. We thank the seminar participants at the Center for Global Sustainability of the University of Maryland for their suggestions and recommendations during the preparation of this draft.

## Author contributions
J.L. and X.S. were responsible for designing the original study and carrying out the analysis. J.L. conducted the entire analysis. D.N. and N.H. contributed to the study design and provided guidance throughout the writing process.

## Competing interests
The authors declare no competing interests.

## Inclusion & Ethics Statement
The research aims to enhance our comprehension of the distribution and accessibility of publicly available electric vehicle (EV) infrastructure in the United States. All authors contributing to this study are located within the United States, obviating the necessity of inviting local researchers. Where feasible, we have endeavored to incorporate both local and national research into our citations, including studies focused on regions such as California and New York City, although such research remains relatively limited at present. Additionally, the study received exemption from the institutional review board (IRB) at the University of Maryland with the number (#2176750-1) under the Exemption category #45CFR46.104(d)(4)(ii).
