## [Peer Review File · Nature Communications]

REVIEWER COMMENTS

Reviewer #1 (Remarks to the Author):

In my opinion, this article is of very high quality. Overall, it is very well written, clear, and precise, and reflects a great deal of work. It looks at disparities in access to charging infrastructure from the angle of income and racial disparities. At a time when universal access to recharging infrastructure is crucial to ensuring the democratization of electric vehicles and decarbonizing the transport sector, a subject like this is both relevant and topical.

The results presented are interesting. Some seem logical, but it's always good to prove such results. Others are more surprising and innovative, such as the differences in access between MUDs and SFDs in urban and rural areas. They open some very interesting new research questions to explore.

My only real concern is about the criterion chosen to study the accessibility of the infrastructure: the proximity to a charging station. By choosing this criterion, I see several possible biases. For example, an area with just one slow charging station in its center will somehow be "rated" higher in terms of accessibility than an area a few kilometers from a charging hub with fast charging stations while it's obviously not - or at least that's the impression I get from reading the paper.

This is why an indicator such as the ratio of inhabitants/number of stations would have seemed more relevant to me (even if it is not free from bias either). However, it should be noted that the authors have made the effort to look at distances not only to the nearest station, but also to the next 5, which makes their indicator more relevant. This effort could be mentioned earlier in the paper (e.g., around line 77-78 when the authors give the criterion they chose), as it makes the paper more interesting.

My following comments are only minor suggestions or questions

- It is clear from reading the paper that the authors have paid attention to possible correlations or collinearities between race and income. However, nowhere is this explicitly stated, so it might be a good idea to write it down, to remove any doubt.

- The authors speak of "rural" and "urban" areas, but do not define these terms. Where is the boundary between the two? For example, is a residential suburb urban or rural?

- On line 109-111, I'm not sure to understand the sentence "African American households experience the least accessibility to public EV infrastructure in rural areas. In urban areas, Black households are the second least accessible group, followed by White households". So black households are the ones with the best access in urban areas? And later you write that you are only talking about "racial accessibility gaps between Black and White households", so why "second least" if there are only two groups? (but maybe it's just my English that is not good enough)

- Figure 1 is too small to be readable in A4 format. 1.a and 1.b should be separated so that they cover the whole width of the page.

- The authors have chosen two types of accessibility gaps, racial and financial. It might be useful to mention a few others and explain why they have been omitted.

- You use the term "accessibility gaps", the term "barriers" seems to me to be used more often and therefore might be better for the article to appear in search engines, but that may just be my impression.

- I have a problem with paragraph 170-175. "Public resources, specifically the public EV infrastructure, should be equitably accessible [...]". I completely agree, but even if they are accessible to the public, are EV infrastructures "public" when private companies decide to install them, own them and operate them? From then on, it's a market and no longer a public good.

In that case, wouldn't the message to be put across be that public EV infrastructure constitutes a critical market of national interest (in the same way as oil, water, electricity, etc.), and that this market should at least be heavily regulated by public authorities to ensure access for all? This wording sends out more or less the same message but would be more difficult to attack.

- Line 305, typo, "SDF units" instead of "SFD".

Reviewer #2 (Remarks to the Author):

Dear Authors,

Authors should consider additional analyses to support the claims made in regressions analysis at the county level to analyze the factors contributing to the racial and income gaps in EV infrastructure accessibility. The paper should clearly outline the research objectives and hypotheses. Need to provide more details on uncertain variables used to assess income and racial disparities in EV infrastructure accessibility. The paper relies on 2021 micro-level data from a large number of households, which is commendable. However, authors should mention the data normalization. A more comprehensive research gap is needed to provide a better context for the proposed study. This suggests that local governments should implement tailored solutions, it would be helpful to provide more specific government policy recommendations based on the existing policies. How can government or policymakers address the disparities identified in the study? Overall, the paper's contribution and organization need improvements.

Response to reviewer #1

Response: We thank the reviewer for the thoughtful comments on the manuscript. Below we have addressed the reviewer's comments in detail and provided the support of additional analyses where needed. We think the manuscript is much improved as a result of reviewer comments.

1. Reviewer 1 comment

Reviewer #1 (Remarks to the Author):

In my opinion, this article is of very high quality. Overall, it is very well written, clear, and precise, and reflects a great deal of work. It looks at disparities in access to charging infrastructure from the angle of income and racial disparities. At a time when universal access to recharging infrastructure is crucial to ensuring the democratization of electric vehicles and decarbonizing the transport sector, a subject like this is both relevant and topical.

The results presented are interesting. Some seem logical, but it's always good to prove such results. Others are more surprising and innovative, such as the differences in access between MUDs and SFDs in urban and rural areas. They open some very interesting new research questions to explore.

My only real concern is about the criterion chosen to study the accessibility of the infrastructure: the proximity to a charging station. By choosing this criterion, I see several possible biases. For example, an area with just one slow charging station in its center will somehow be "rated" higher in terms of accessibility than an area a few kilometers from a charging hub with fast charging stations while it's obviously not - or at least that's the impression I get from reading the paper.

This is why an indicator such as the ratio of inhabitants/number of stations would have seemed more relevant to me (even if it is not free from bias either). However, it should be noted that the authors have made the effort to look at distances not only to the nearest station, but also to the next 5, which makes their indicator more relevant. This effort could be mentioned earlier in the paper (e.g., around line 77-78 when the authors give the criterion they chose), as it makes the paper more interesting.

Response: We genuinely value the thoughtful insights provided by the reviewer, particularly their concern regarding our choice of studying infrastructure accessibility concerning proximity to charging stations. As the reviewer notes, we expand our analysis beyond just the nearest station to encompassing distances to the subsequent five stations. We wholeheartedly concur with the reviewer's observations regarding the "friction" in accessibility that might occur between Level 2 charging stations and DC fast charging stations.

In response to the reviewer's comments, we have taken three distinct approaches to enhance our analysis; these results are provided in the Supplementary Note 1 in the Supplementary Information (SI). Additionally, we have revised our manuscript to reflect results from these analyses. Our aim is to ensure that each of these approaches contributes to a comprehensive and robust examination, ultimately addressing the reviewer's concerns and providing confidence in the validity of our findings. The reviewer can also review the revisions below:

Manuscript (lines 150-160):

“Public EV infrastructure typically contains level 2 and Direct Current fast charging (DC fast charging) (US Department of Transportation, 2023). In this paper, we focus on infrastructure accessibility, specifically the proximity to charging stations. Potential bias may arise from consumers’ preferences for DC fast charging stations over level 2 charging stations. To account for this, we undertake three additional robustness analyses to bolster our primary analysis, all of which are detailed in Supplementary Note 1 within the Supplementary Information (SI). One of our analyses introduces an index that accounts for users’ preferences across various charging station types by assigning distinct weights to DC fast charging stations, another by examining the ratio of DC fast charging stations in our core analysis, and finally incorporates the analysis using the Inhabitant-to-Station Ratio through a two-tiered approach. The results consistently align with the overarching trends observed in our core assessment.”

Supplementary Note 1 (SI pages 31-38):

This supplementary note presents the robustness analysis for our core assessment of accessibility to EV charging stations. Our baseline analysis is grounded in the calculation of the shortest distance between households and the closest five public charging stations. However, we are fully aware that this seemingly straightforward measure may incorporate bias stemming from users’ varying preferences for DC fast charging stations versus Level 2 charging stations versus. Consequently, we have implemented three additional distinct analyses to assess the robustness of our measurements.

Approach 1: In our examination of the nearest five public charging stations, we find that a substantial number of DC fast charging stations are included.

In our examination of the nearest five public charging stations, it is significant to emphasize that a substantial number of DC fast charging stations are included in the nearest five stations analysis. We find that households with at least one DC fast charging station in the five station subset ranges from around 18% upwards to 85% depending on the state (see Supplementary Table S8), with a national average of 40%. Given the substantial representation of DC fast charging stations in our core assessment of accessibility, even with state-to-state variability, any potential bias in our findings will be small.

Supplementary Table S8. Count of Households with DC Fast Charging Stations Among the Five Nearest Public Charging Stations.

state	Zero Fast	One fast station	Two fast stations	Three fast stations	Four fast stations	Five fast stations	Total number of households that have at least one fast charging station among the five nearest stations	Ratio of households with DC fast charging
AK	168844	79559	48110	28587	2357		158613	48%
AL	1073298	311674	287037	237041	67944	98108	1001804	48%
AR	845256	228057	80042	75123	12370	4170	399762	32%
AZ	1759040	370972	241489	109415	21885	6940	750701	30%
CA	9453366	2757110	1579831	921729	354337	242392	5855399	38%
CO	1413284	471319	251003	117034	56407	24204	919967	39%
CT	938128	305273	116271	66275	31137	119	519075	36%

DE	188853	110055	59084	26834	10020	66	206059	52%
DC	233554	50263	5133				55396	19%
FL	4941155	1715362	1215525	524567	254193	172287	3881934	44%
GA	2264335	670545	553365	319167	75045	51766	1669888	42%
HI	285752	63495	57452	24148	10454	8977	164526	37%
IA	497330	224949	224185	153001	107545	62463	772143	61%
ID	326110	113298	96440	62612	59226	11	331587	50%
IL	3282172	877274	660341	307468	84456	44835	1974374	38%
IN	1787894	591452	279698	152976	104738	80183	1209047	40%
KS	766921	126051	112348	25973	12851	9764	286987	27%
KY	1243437	276831	153637	63073	10410	9997	513948	29%
LA	1259950	662204	243343	112393	27358	24165	1069463	46%
MA	2284941	331264	121155	37660	16687	2657	509423	18%
ME	467056	97134	50273	27230	2595	2057	179289	28%
MD	1281192	475679	254842	162868	57574	17439	968402	43%
MI	2455623	669141	518063	310436	127653	47726	1673019	41%
MN	1189970	341660	297900	128860	114279	45091	927790	44%
MO	1173795	264768	394293	343080	140650	68413	1211204	51%
MS	526310	287185	283604	15482			586271	53%
MT	169595	163614	42398	27662	25597	16404	275675	62%
NC	2666368	964851	492054	200596	62556	23098	1743155	40%
ND	108429	68983	51190	36807	25475	13887	196342	64%
NE	397436	182987	92811	31195	17728	9880	334601	46%
NH	395396	100395	62492	16479	11494	10041	200901	34%
NJ	1761029	556205	481641	258072	119028	92868	1507814	46%
NM	538111	60591	59442	57636	19843	10139	207651	28%
NV	662470	277868	185647	91979	36604	32547	624645	49%
NY	6186854	1010291	548767	207064	76793	26188	1869103	23%
OH	3602964	680177	371044	142988	30080	6841	1231130	25%
OK	230767	150534	231701	286134	316191	399887	1384447	86%
OR	870384	318065	183796	115192	58921	49274	725248	45%
PA	3129526	948956	491919	258488	93898	52554	1845815	37%
RI	339207	57281	33949	11596	5252	1218	109296	24%
SC	1232054	429490	244280	98358	17051	1241	790420	39%
SD	111037	79691	58324	32123	31143	26627	227908	67%
TN	1606275	498303	244931	192892	54526	10268	1000920	38%
TX	7245189	2561381	1802681	624394	284344	64645	5337445	42%
UT	679294	181969	51879	32389	27596	20497	314330	32%
VA	1667665	656626	511588	242847	111970	26979	1550010	48%
VT	217149	60802	36272	11110	1492	242	109918	34%
WA	1618196	505282	311415	187811	98608	36858	1139974	41%
WI	1580612	799252	484481	57001	12214	5011	1357959	46%
WV	398316	273629	66252	21885	14322	6451	382539	49%
WY	52626	47978	46474	42696	35289	6902	179339	77%

Note: The “Zero Fast” column indicates the count of households that among the five nearest stations, there are no DC fast charging stations available. The “One fast” column represents the count of households that have only one DC fast charging station among the five nearest stations. The “Two fast stations” column reflects the count of households with two DC fast charging stations among the five nearest stations. The “Total number of households that have at least one fast charging station among the five nearest stations” columns are the sum of columns of “One fast station”, “Two fast stations”, “Three fast stations”, “Four fast stations”, and “Five fast stations”. The “Ratio of households with DC fast charging” column is computed by dividing the “Total number of households that have at least one fast charging station among the five nearest stations” by the entire number of households.

Approach 2: We introduce an alternative index that captures users’ preferences for various types of charging stations. Our findings show it consistently aligns with the overarching trends observed in our core assessment.

To account for potential user preferences for DC fast charging stations compared to Level 2 chargers, we conducted a new analysis using an alternative index we introduce. Initially, we separated regular charging, typically Level 2 charging, from DC fast charging, and computed the shortest distances from household addresses to their respective Level 2 (D_s) and DC fast charging (D_f) stations, respectively. Here, we combine these two shortest distances, D_s and D_f , into a unified indicator, assigning distinct weights to the α , using the algorithm:

Step 1: Calculate the distance from each resident's address to the nearest Level 2 charging station (D_s) and Fast DC charging station (D_f) separately, for each of the 121 million households in the United States;

Step 2: Compare each household's D_s and D_f , and derive the final accessibility index using the following heuristics:

Rule 1: If $D_s \leq \text{Walking distance}$, & $D_f \leq \text{walking distance}^*$, accessibility = $\min \{D_s, D_f\}$

Rule 2: If $D_s > \text{Walking distance}$, & $D_f \leq \text{walking distance}^*$, accessibility = $\min \{D_s, D_f\}$

Rule 3: If $D_s \leq \text{Walking distance}$, & $D_f > \text{walking distance}^*$, accessibility = $\min \{D_s, D_f\}$

Rule 4: If $D_s > \text{Walking distance}$, & $D_f > \text{walking distance}^*$, accessibility = $\min \{D_s, \alpha D_f\}$, where $\alpha < 1$, and α is in the range of (0.5,1)

*walking distance: we have chosen 0.7 mile as the walking distance, which is about 15 min, from the literature (Yang and Diez-Roux, 2012).

Our heuristics are predicated on the assumption that when residents have access to charging stations within walking distance from their homes, the difference in preference between level 2 and DC fast charging becomes minimal. Residents can leave their cars charging and easily return home to engage in other activities. When the stations are situated beyond walking distance, it becomes more challenging for individuals to arrange their activities during the charging period, thereby increasing the attractiveness of DC fast charging. However, accurately quantifying the preference for fast charging over Level 2 charging in this scenario presents a significant challenge. Consequently, we conduct a sensitivity analysis by multiplying the shortest distance to DC fast charging stations with a weight, denoted as α , and allow α to range from 0.5 to 1.

Depending on the specific value of α (s) we adopt, we can derive a range of EV charging accessibility indexes. When α equals 0, it signifies the exclusive selection of DC fast charging stations, with no consideration for Level 2 charging stations, even if a Level 2 station is much closer than a distant fast charging station. However, such a scenario is unlikely as people can typically manage their activities during charging. Therefore, we start α at 0.5. Conversely, when α equals 1, both Level 2 and DC fast charging stations are assigned equal weight (our core analysis). We conduct a sensitivity analysis to bound our findings.

We re-run our LOESS analysis and examine income and racial disparities within the spectrum of EV accessibility using our index. Remarkably, our sensitivity analysis demonstrates consistent trends across the range of α 's we tested (illustrated in Supplementary Figure S22). These results are with our core analytic results. Similarly, our examination of the geographical distribution of income and racial disparities (as demonstrated in Supplementary Figure S23), even when

employing the most aggressive weight α of 0.5, still yields results that closely mirror our findings outlined in the manuscript. Thus, our conclusions in the manuscript are robust.

Supplementary Figure S22: LOESS regression curves among different racial groups by average distance to nearest EV infrastructure in rural and urban areas based on the weights we assigned to the fast-charging station. alpha0.5 means we assign 0.5 weight to fast charging. The top panel of ten subplots represents results for urban households, while the bottom panel of ten subplots represents results for rural households.

a. State income gap rural

b. State income gap urban

c. County income gap rural

d. County income gap urban

e. State racial gap rural

f. State racial gap urban

g. County racial gap rural

h. County racial gap urban

Supplementary Figure S23: Equity assessment through accessibility gaps at state and county level using alpha (α) 0.5. a-d: accessibility gaps between LMI and non-LMI households by their location at state and county; e-h: accessibility gaps between black and white groups by their location at state and county. The magenta color scheme shows greater gaps, and the green color scheme shows less gaps. Gray color means no data available or beyond the defined minimum and maximum thresholds of -300 and 300, respectively.

Approach 3: We conduct the analysis using the Inhabitant-to-Station Ratio as suggested by a reviewer.

To strike a balance between representing income and racial distributions while ensuring access to sufficient EV charging infrastructure, we apply a two-tiered approach to calculate the Inhabitant-to-Station Ratio (ISR) at the zip code and the census tract levels.

1. At the zip code level, our choice is rooted in the fact that our rural/urban categorization, based on U.S. Census data, operates at this level. Consequently, we can segregate zip codes into rural and urban categories and calculate the ISR accordingly. To classify zip codes into LMI (Low to Moderate Income) and non-LMI groups, as well as Black and White demographic categories, we followed these steps: We utilize median household income at the county level as the threshold. If the median household income at the zip code level falls below this threshold, we categorize it as LMI; if it surpasses the threshold, it is labeled as non-LMI. For racial categorization, if the percentage of the Black population exceeds the percentage of any single racial group (including White, Hispanic, Asian, Hawaiian, Native American, etc.), the zip code is classified as Black-dominant, and vice versa for White-dominant zip codes. Subsequently, we computed the ISR gaps between LMI and non-LMI and Black and White groups, aggregating the means of ISRs at the state level, mirroring the format presented in Figure 2 in our manuscript. Please refer to Supplemental Figure S24 for the ISR at the state level based on zip code information.
2. Moving to the census tract level, these areas are smaller than zip codes and represent the lowest jurisdictional level for which we have comprehensive data from U.S. census, including the number of households, EV charging stations, racial demographics, median household income, and more. We applied a similar methodology to our zip code approach to calculate the ISR gap, with one additional step of assigning a rural/urban indicator to each tract. This step is necessary since rural/urban classification is only available at the zip code level. We use a 2010 ZCTA to Census Tract Relationship File to link zip codes to census tracts. In cases where a tract spanned multiple zip codes, we distributed the tract's population among the relevant zip codes and linked it to the one with the largest population share. Please refer to Supplemental Figure S25 for the ISR gaps at the state level based on census tract information.

There are some concerns regarding the utilization of the ISR that we note. First, it is essential to acknowledge that there will be varying outcomes from the ISR methods across the two levels of study we have adopted. Second, we lack information for certain states when employing ISR methods; this issue becomes particularly prominent in the analysis of racial disparities. This is primarily attributable to the inherent trade-off between granularity and aggregation. 1) When opting for the tract level, a significant number of tracts without public EV charging stations end up being excluded. As a result, we are left with only 16,203 tracts out of 85,187. Subsequently, when we aggregate our analysis to the state level, some states, particularly those in rural areas, may not have enough either LMI or non-LMI tracts to facilitate a meaningful comparison. In fact, in certain cases, we may be forced to exclude an entire state (e.g., state Louisiana) for the income gap analysis, and most of the states in the racial gap analysis, because there are states that do not have Black-dominant tracts. 2) If we decide to use zip code data instead, we risk losing the ability

to capture the intricate dynamics of income and racial distribution due to the broader geographical scale of zip codes compared to tracts. This issue becomes more pronounced in the context of racial analysis. The point of this is that we expect our original results derived using the individual household level will yield more robust results when contrasted with higher-level aggregation analyses. Nonetheless, for those geographies that are available, we find consistent results to our base analysis.

Supplementary Figure S24. Equity assessment through Inhabitant-to-Station Ratio (ISR) gaps at state level based on the *zipcode*. a and b: ISR gaps between LMI and non-LMI households by their location at state level; c and d: ISR gaps between Black and White groups by their location at state level. The magenta color scheme shows LMI and Black households experienced larger ISR gaps (or barriers) compared to non-LMI and White households, respectively. Vice versa, the green color suggests the opposite. Gray color means no data available.

Supplementary Figure S25. Equity assessment through Inhabitant-to-Station Ratio (ISR) gaps at state level based on the *census tract*. a and b: ISR gaps between LMI and non-LMI households by their location at state level; c and d: ISR gaps between Black and White groups by their location at state level. The magenta color scheme shows LMI and Black households experienced larger ISR gaps (or barriers) compared to non-LMI and White households, respectively. Vice versa, the green color suggests the opposite. Gray color means no data available.

2. Reviewer 1 comment

My following comments are only minor suggestions or questions

- It is clear from reading the paper that the authors have paid attention to possible correlations or collinearities between race and income. However, nowhere is this explicitly stated, so it might be a good idea to write it down, to remove any doubt.

Response: Thank you. We did pay attention to the possible correlation between race and income. To enhance the manuscript's clarity, we have incorporated a new paragraph in the discussion section in the manuscript (lines 491-496),

“While our paper primarily delves into two distinct accessibility gaps, one associated with income disparities and the other centered around racial disparities, we are acutely aware of the potential correlation between race and income. Our county-level regression analysis in section *Potential contributors to the social disparities in public EV infrastructure* illustrates that the income gap significantly influences the accessibility gap in the context of racial analysis.”

3. Reviewer 1 comment

- The authors speak of "rural" and "urban" areas, but do not define these terms. Where is the boundary between the two? For example, is a residential suburb urban or rural?

Response: Our definition of rural and urban is based on the 2010 Census Urban and Rural Classification and Urban Area Criteria from the US Census Bureau. This definition was in use when we started the manuscript in 2022, as the 2020 version was not available at that time. According to this definition, all cities, towns, and suburbs are included in our urban category, while the remaining regions are classified as rural. To ensure the robustness of our definition, we conducted a cross validation by using the more detailed breakdown provided by the Department of Education, National Center for Education Statistics (NCES). The NCES classifies all territories in the U.S. into four types: rural, town, suburban, and city (Geverdt, 2019). Upon comparing the two definitions, the urban areas identified in our paper perfectly match the town, suburban, and city categories from the NCES definition (see the maps below, which is marked as supplementary Figure S21).

To clarify the definition in our manuscript, we have revised the sentences that describe our urban and rural definitions to enhance understanding (lines 533-541),

“1) **Rural and urban definition.** We utilized the 2010 Census Urban and Rural Classification and Urban Area Criteria from the US Census Bureau. Our study includes both the urbanized areas (UAs, 50,000 or more people) and Urban Clusters (UCs, between 2,500 and 50,000) as the urban areas. This definition encompasses residents from cities, suburbs, and towns, aligning seamlessly with the town, suburban, and city categories suggested by the Department of Education, National Center for Education Statistics (NCES) definition (refer to Supplementary Figure S21b) (Geverdt, 2019). We compiled the 2010 Urban Area list file from Census based on the urban area census code (UACE) and converted all the UACE to the zipcode level using the 2010 Urban Area to ZIP Code Tabulation Area (ZCTA) Relationship File.”

Supplementary Figure S21. Comparison of the urban and rural definitions from the Census to the National Center for Education Statistics (NCES). a.2020 Census Urban Areas. b. 2021 NCES Locale Classification.

4. Reviewer 1 comment

- On line 109-111, I'm not sure to understand the sentence "African American households experience the least accessibility to public EV infrastructure in rural areas. In urban areas, Black households are the second least accessible group, followed by White households". So black households are the ones with the best access in urban areas? And later you write that you are only talking about "racial accessibility gaps between Black and White households", so why "second least" if there are only two groups? (but maybe it's just my English that is not good enough)

Response: We appreciate the insightful feedback from the reviewer. Your point is absolutely valid and we have clarified the text. To explain, when conducting the locally weighted scatterplot smoothing (LOESS) regression, which is the first research design, we consider all four major racial groups (White, Black, Hispanic, and Asian). As a result, when discussing Figure 1, which portrays

these four racial groups with distinct lines. However, as we progress through the remainder of the research designs, our objective is to concentrate on the accessibility gap between specific racial groups. For this purpose, we choose to narrow our focus to two specific groups: White and Black households. We have also offered a rationale for this selective focus, as outlined in lines 193-197.

“In this analysis, we focus on the racial accessibility gaps between Black and White households instead of all types of racial groups. Our interest in the spatial differences between Black and White households stems from the persisting disparities and inequities faced in accessing social resources, including transportation infrastructure, health, education, and more. (Battelle, 2000; Maness et al., 2021; Oliver and Shapiro, 2013)”

Once again, we extend our gratitude to the reviewer for raising this important point. We are committed to enhancing the clarity of our manuscript to eliminate any potential confusion. We have revised the manuscript text (lines 131-138) as,

“Using locally weighted scatterplot smoothing (LOESS) regression and US population data, we find a consistent (nonlinear) positive relationship between income and public EV infrastructure accessibility across four major racial or ethnic groups (Figure 1a). In both urban and rural areas, as household income increases, access to the public charging station(s) also increases (the distance to the charging station decreases). African American households experience the least accessibility to public EV infrastructure in rural areas conditioning on income, compared to White, Hispanic, and Asian households. In urban areas, Black households are the second least accessible group, followed by White households.”

5. Reviewer 1 comment

- Figure 1 is too small to be readable in A4 format. 1.a and 1.b should be separated so that they cover the whole width of the page.

Response: Thank you. We have revised Figure 1 to enhance its readability in the manuscript. The updated figure is provided below:

Figure 1. a. LOESS regression curves among different racial groups by average distance among N-nearest EV infrastructure in rural and urban areas. b. Increased rates (%) between Nth and (N-1)th nearest EV infrastructure accessibility among different racial groups and income groups in urban and rural areas. EV public infrastructure accessibility showing the LOESS regression curves by \$100,000 interval of the entire sample ($n = 11,8682,791$). The LOESS regression curve, represented by the solid line in the center, shows the predictive value, while the shadow around it represents the 95% confidence interval.

6. Reviewer 1 comment

- The authors have chosen two types of accessibility gaps, racial and financial. It might be useful to mention a few others and explain why they have been omitted.

Response: There are indeed many other dimensions of inequality but income and race are two fundamental, rooted problems that capture access to EV infrastructure. Income is a fundamental metric to measure people's economic ability and social class. Racial inequality and discrimination are a structural problem in the U.S. A significant literature focuses on these two factors (Barbose et al., 2020; Borenstein and Davis, 2016; Brown, 2022; Davis, 2019; Gao and Zhou, 2022; Sunter et al., 2019). More practically, income and race are two key indicators we can observe at the household level (nationwide). However, the reviewer's comment prompted us to explore incorporating the Electric Vehicle (EV) Charging Justice40 Map Tool as a third dimension of accessibility gap. This addition is useful and certainly increases the relevance and rigor of the analysis. We have made significant revisions to our manuscript and provided a comprehensive

explanation in **Supplementary Note 2** in the Supplementary Information (SI). We extend an invitation to the reviewer to evaluate the following sections:

Manuscript (lines 182-188):

“To evaluate equity at the nation-wide county level, we quantify two types of accessibility gaps: one related to income disparities and the other centered around racial disparities as our core dimensions of accessibility gaps. We have chosen income and race are because they represent two structurally rooted problems. Income is a fundamental metric to measure people’s economic ability and social class. Racial inequality and discrimination is another deeply rooted, systemic problem in the U.S. A vast literature focuses on these two factors (Barbose et al., 2020; Borenstein and Davis, 2016; Brown, 2022; Davis, 2019; Gao and Zhou, 2022; Sunter et al., 2019). More practically, income and race are two key indicators we can observe at the household level nation-wide in our dataset.

However, we recognize the existence of various dimensions related to accessibility gaps, encompassing factors like vulnerability, social and economic disadvantage, and educational attainment, among others. In our pursuit of an exhaustive analysis on the accessibility gaps from income and racial dimensions, we conducted an additional investigation by incorporating the Electric Vehicle (EV) Charging Justice40 Map Tool, jointly proposed by the Department of Energy and Department of Transportation. This tool, designed to identify Disadvantaged Communities (DAC) at the census tract level, offers an alternative perspective on accessibility gaps (refer to Supplementary Note 2) between DAC and non-DAC. Our examination revealed strong correlations between our racial accessibility gap and the DAC accessibility gap, as well as between our income accessibility gap and the DAC accessibility gap, both at the county and state levels. It is crucial to underscore that our analyses of income and racial disparities are firmly grounded in individual household data, providing a finer-grained understanding compared to analyses conducted using demographic data at the broader census tract level. This heightened level of detail enhances our confidence in the robustness of our findings, particularly when comparing them with the DAC accessibility gap. We note that our analysis suggest that the Justice40 tool may be too blunt to capture important variations in EV infrastructure access.”

Supplementary Note 2 (SI pages 39-40):

This supplementary note introduces an additional dimension to the accessibility gap by directly incorporating the concept of Disadvantaged Communities (DAC) as defined by the Electric Vehicle (EV) Charging Justice40 Map Tool.

The EV Charging Justice40 Map Tool, a collaborative effort between the Department of Energy and the Department of Transportation, encompasses 36 and 22 indicators that define Disadvantaged Communities (DAC) at the census tract level (see Supplementary Figure S26). This tool offers a comprehensive perspective on DACs. To integrate this dimension into our analysis, we assigned each household a binary variable denoting its DAC status (1 for DAC, 0 for non-DAC). Subsequently, we calculated the accessibility gap between DAC and non-DAC households at both county and state levels.

This expanded analysis allowed us to explore correlations between our racial accessibility gap and the DAC accessibility gap, as well as between our income accessibility gap and the DAC

accessibility gap, both within counties and across states. For detailed correlation results, please refer to Supplemental Table S9. We highlight a significant limitation associated with the use of the DAC classification. We've noted that many counties are categorized as either entirely non-DAC or entirely DAC, despite the DAC classification being based on the census tract level. Consequently, this results in a significant reduction in the number of counties included in our final county-level correlation analysis. To provide context, there are 860 counties that do not contain any DAC census tracts, and 273 counties exclusively consist of DAC census tracts. In such cases, where counties lack diversity in their DAC status, we encounter limitations in our ability to compare accessibility gaps between DAC and non-DAC groups within these counties. This phenomenon contributes to the observed decrease in the strength of correlation at the county level compared to the state level.

Supplementary Table S9 Correlation Coefficients Between the Racial Accessibility Gap and the DAC Accessibility Gap, and Income Accessibility Gap and the DAC Accessibility Gap.

Correlation	Income accessibility gap and DAC accessibility gap	Racial accessibility gap and DAC accessibility gap
State	0.43	0.27
County	0.31	0.21

It's crucial to underscore that our dataset operates at the household level, affording us the capability to delve into fine-grained details regarding income and race. Any level of aggregation above household will necessarily be less rigorous. Other metrics, e.g., "Inhabitant-to-Station Ratio" do not capture individual-level variation. Our study's granularity (i.e., the household) instills confidence in the robustness of our results when juxtaposed with the DAC accessibility gap. We note that our analysis suggest that the Justice40 tool may be too blunt to capture important variations in EV infrastructure access.

Supplementary Figure S26. DOE/DOT's interim definition for disadvantaged communities (DACs)

7. Reviewer 1 comment

- You use the term “accessibility gaps”, the term “barriers” seems to me to be used more often and therefore might be better for the article to appear in search engines, but that may just be my impression.

Response: The concept of accessibility has its origins in the fields of transportation and urban planning, where it refers to the ease of reaching various destinations. Factors such as available transportation options, surrounding land-use patterns, and socio-economic characteristics, among others, can play a role in determining the levels of accessibility (Geurs and van Wee, 2004; Handy and Niemeier, 1997; Levine et al., 2019). Given that our paper revolves around electric vehicles (EVs) and EV infrastructure, it is fitting for us to adopt this term.

However, we acknowledge that the term “barriers” is more commonly employed in other social science disciplines. We use both “gap” and “barrier”, but “accessibility gap” remains our major term. We have also included a paragraph (lines 89-99) to briefly address this transition while retaining the core concept of accessibility, as EVs squarely fall within the domain of transportation,

“We use a detailed 2021 national dataset, consisting of micro-level data from 121 million households across 51 states to estimate the social disparity in public EV infrastructure accessibility. The concept of accessibility has its origins in the fields of transportation and urban planning, where it refers to the ease of reaching various destinations (Geurs and van Wee, 2004; Handy and Niemeier, 1997; Levine et al., 2019). It has since evolved and extended its reach into the broader domain of social science, where the term “barrier” has been more frequently adopted.”

8. Reviewer 1 comment

- I have a problem with paragraph 170-175. "Public resources, specifically the public EV infrastructure, should be equitably accessible [...]". I completely agree, but even if they are accessible to the public, are EV infrastructures "public" when private companies decide to install them, own them and operate them? From then on, it's a market and no longer a public good. In that case, wouldn't the message to be put across be that public EV infrastructure constitutes a critical market of national interest (in the same way as oil, water, electricity, etc.), and that this market should at least be heavily regulated by public authorities to ensure access for all? This wording sends out more or less the same message but would be more difficult to attack.

Response: We thank the reviewer for this excellent comment. The reviewer raises a very important perspective: that public EV infrastructure constitutes a critical market of national interest, and that this market should at regulated by public authorities to ensure equitable access. We conducted additional research on how the current policy, both at the federal and subnational, ensures the equitable access to the public EV infrastructure, and what additional efforts the public authorities can do in the discussion section. For our comprehensive policy analysis, please refer to the Supplementary Note 3. The reviewer can also check the Supplementary Note 3 and the revised manuscript as follows:

Manuscript (lines 433-448):

“A crucial aspect of this transition is the strategic development of EV charging stations in economically disadvantaged regions, ensuring that the benefits of electric transportation are accessible to all segments of society. We conducted an extensive analysis of EV charging regulations and incentives at the federal and state government levels (see Supplementary Note 3 for the comprehensive database). We found that there are only 26 existing regulations aiming to promote equitable access to electric vehicle charging infrastructure implemented in 16 states. In contrast, incentives are more common in states across the US. This indicated a clear gap of regulations from the state government which suggest that subnational government should implement tailored solution. Given that public EV infrastructure is of significant national interest, it is crucial for this market to be well-regulated by public authorities to ensure access for all.”

Supplementary Note 3. EV Charging Regulations and Policies (SI pages 41-46):

The current federal and some state policies related to EV public infrastructure emphasize increasing the availability of charging stations and providing incentives for their installation. However, states that have policies mostly focus on broader installation, with a few states placing equity considerations front and center. It is crucial to navigate the development of EV charging stations in disadvantaged regions, ensuring that the benefits of electric transportation are accessible to all members of society. We conducted a comprehensive analysis of EV charging regulations and incentives from the federal and state government. We found that there are only 26 existing regulations aiming to promote equitable access to electric vehicle charging infrastructure implemented in 16 states. In contrast, incentives are more common in states across the US. It is evident that the United States currently lacks well-defined regulations and rules to promote equitable access to electric vehicle charging infrastructure (Smart Cities Dive, 2022). This indicated a clear gap of regulations from the state government which suggest that subnational government should implement tailored solution.

Regulations/Rules

At the federal level, one of the most significant motivators of equity and access within EV charging development is the Justice40 program, which sets a goal of providing 40% of the benefits of relevant programs to disadvantaged communities (DACs). A major federal program under the umbrella of Justice40 is the National Electric Vehicle Infrastructure (NEVI) Formula Program, which requires states to submit comprehensive plans in order to receive crucial funding for deploying EV charging around important highways (Patterson et al., 2023). In its administration of this program, the federal government has both directly and indirectly prioritized equity concerns. First, states are required to touch on equity in their NEVI plans, although there is an absence of specific requirements in terms of implementation. The Federal Highway Administration implemented NEVI standards and requirements to ensure federally funded charging stations meet a certain level of accessibility, including charging network connectivity and publicly available price and availability information (“National Electric Vehicle Infrastructure Standards and Requirements,” 2023).

At the state level, there is significant variation in implementation of NEVI guidance as well as local EV infrastructure policy (Patterson et al., 2023). To start, states have taken an array of strategies to plan for Justice40 compliance. Most states that submitted NEVI plans use the Electric Vehicle Charging Justice40 Map created from the Department of Transportation and Department of Energy’s joint interim definition of DACs (Lyons, n.d.; Patterson et al., 2023). In addition, a number of states have integrated tools that utilize more specific data available to states that may not be consistent enough across the country to be used in the federal tool. These generally utilize multiple data sets to measure the climate and pollution burdens faced by each census tract or locality, considering factors like air quality, health risks, and socioeconomic demographics. States also do not have a consistent definition of what a “benefit” looks like to a DAC, and to some degree have used their NEVI plans to describe these benefits with varying degrees of specificity. State defined benefits have included air quality benefits most prevalently, as well as workforce participation and economic development from tourism (Patterson et al., 2023). Finally, some states conducted community outreach to formulate their NEVI plans. Successful approaches ranged from social media campaigns and targeted newspaper ads to regional workshops. However, one report states these activities “were lacking involvement from groups that represented low-income communities, communities of color, and environmental justice communities” (Patterson et al., 2023). While most states did engage with the public, some only held private meetings; states also varied in the degree to which they disclosed the specifics of their outreach and incorporated feedback in NEVI plan formulation (Patterson et al., 2023).

Outside of NEVI, there has been little coordination at the state level to facilitate equitable EV regulations. Of the regulations that are in place, most are at best indirectly linked to equity and access. In our database, we are only able to find 16 states implemented regulations to ensure the equitable accessibility, see Supplementary Table S10.

Supplementary Table S10. A List of Regulations Among 16 States.

State	Policy Title	Law referred to	Policy related to equity	Targeted Communities	Date Started	Links/Documents
CA	Electric Vehicle (EV) Charging Station Assessment	California Public Resources Code 25229	In addition, the assessment must analyze the existing and future infrastructure needs across California, including in low-income communities .	Low-income		http://leginfo.ca.gov/
CA	Electric Vehicle (EV) Charging Station Uptime Reporting Standards	Assembly Bill 2061, 2022	The assessment must include considerations for equitable access to EV charging stations in low-, moderate-, and high-income communities .	Low-income	2022	https://leginfo.ca.gov/faces/home.xhtml
CA	Zero Electric Vehicle (ZEV) Office Authorization and Equity Assessment	Senate Bill 1251, 2022	1. Improve access to ZEVs, supporting infrastructure, and ZEV transportation options in low-income, disadvantaged, and underserved communities . 2. Reduce pollution from transportation in low-income, disadvantaged, and underserved communities .	low-income, disadvantaged, and underserved communities	2022	https://leginfo.ca.gov/faces/billNavClient.xhtml?bill_id=202120220SB1251
CA	Zero Emission Vehicle (ZEV) Initiative	California Health and Safety Code 44258.4	1. In consultation with the State Energy Resources Conservation and Development Commission, CARB prepared a funding plan (PDF) that includes a market and technology assessment, assessments of existing zero and near-zero emission funding programs, and programs that increase access to disadvantaged, low-income, and moderate-income communities and consumers. 2. Potential programs under the initiative include those involving innovative financing, car sharing, charging infrastructure in multi-unit dwellings located in disadvantaged communities , public transit, and agricultural vanpool programs.	Disadvantaged, low-income, moderate-income communities, and Multiunit Dwellings		http://www.oal.ca.gov/
CA	Zero Emission Vehicle (ZEV) Promotion Plan	Executive Order B-16, 2012, Executive Order B-48, 2018, and Executive Orders N-19-19, 2019	Update the 2016 ZEV Action plan, with a focus on low income and disadvantaged communities .	Low income and disadvantaged communities	2012	https://www.gov.ca.gov/category/executive-orders/
CA	Electric Vehicle (EV) Pilot Programs	Public Utilities Code 740.13-740.14	Priority must be given to locations in disadvantaged communities , as defined by the California Environmental Protection Agency.	Disadvantaged communities	Sep-20	https://vacleancities.org/mid-atlantic-electrification-partnership/
HI	Electric Vehicle (EV) Charging Station Rebate Program Authorization	Hawaii Revised Statutes 243-3.5, 269-72 and 269-73	The PUC must prioritize rebate awards for EV charging stations that are publicly available; serve fuel cell electric vehicle fleets; serve multiple tenants, employees, or customers; support tourism; or serve low- or moderate-income or environmental justice communities .	Low- or moderate-income or environmental justice communities		https://www.capitol.hawaii.gov/
IL	Electric Vehicle (EV) Charging Station Building Standards for Residential Developments	Senate Bill 40, 2023	1. New single-family homes or small multifamily dwellings that qualify as affordable housing must have one EV-capable parking space per dwelling. 2. Building permits for affordable housing single-family and multifamily developments must meet the following benchmarks to support the development of EV-capable parking spaces.	Affordable Housing	2023	https://www.ilga.gov/legislation/billstatus.asp?DocNum=0040&GAID=17&GA=103&DocTypeID=SB&LegID=142908&SessionID=112
IL	Regional Electric Vehicle (REV) Midwest Plan	REV Midwest Partnership Announcement	Identify historically disadvantaged communities for equitable EV charging station development and EV adoption.	Disadvantaged communities		https://www.michigan.gov/whitmer/0,9309,7-387-90499_90640-569470--00.html
IN	Regional Electric Vehicle (REV) Midwest Plan	REV Midwest Partnership Announcement	Identify historically disadvantaged communities for equitable EV charging station development and EV adoption.	Disadvantaged communities		https://www.michigan.gov/whitmer/0,9309,7-387-90499_90640-569470--00.html

IN	Utility Electric Vehicle (EV) Charging Station Pilot Program Authorization	Indiana Code 8-1-43	Utilities must also include plans to install EV charging stations in underserved and diverse communities .	Underserved and Diverse communities		http://www.in.gov/legislative/ic/code/
MD	Utility Electric Vehicle (EV) Charging Station Deployment Authorization and Requirements	House Bill 834, 2023, Public Service Commission Order 88997 – Case No. 9478, and Maryland Statutes Public Utilities Code 7-901 through 7-905	Beginning October 1, 2023, IOUs that participate in the PSC EV Pilot Program may install Level 2 EV charging stations at multi-unit dwellings in underserved communities .	Underserved Communities	2023	https://vacleancities.org/mid-atlantic-electrification-partnership/
MD	Mid-Atlantic Region Electric Vehicle (EV) Support	MAEP Website	Participating States commit to creating a regional network of EVs and EV charging stations that will make it possible to seamlessly operate light-, medium-, and heavy-duty EVs across transportation corridors and in low-income communities .	Low-income	Sep-20	https://mgaleg.maryland.gov/mgawebsite/Legislation/Details/hb0834?ys=2023RS
MA	Electric Vehicle (EV) Charging Infrastructure Deployment Support	Session Law Chapter 179, Section 81, 2022	Opportunities for EV charging stations in urban, suburban, rural, and low- and moderate-income areas .	Low- and moderate-income areas, and Rural	2022	https://malegislature.gov/Bills/192/H5060
MI	Regional Electric Vehicle (REV) Midwest Plan	REV Midwest Partnership Announcement	Identify historically disadvantaged communities for equitable EV charging station development and EV adoption.	Disadvantaged communities		https://www.michigan.gov/whitmer/0,9309,7-387-90499_90640-569470--00.html
MN	Regional Electric Vehicle (REV) Midwest Plan	REV Midwest Partnership Announcement	Identify historically disadvantaged communities for equitable EV charging station development and EV adoption.	Disadvantaged communities		https://www.michigan.gov/whitmer/0,9309,7-387-90499_90640-569470--00.html
NY	Zero Emission Vehicle (ZEV) Requirements	Senate Bill 7788, 2022	Strategies to accelerate deployment of affordable ZEV infrastructure that serves low-income and disadvantaged communities .	Low-income and disadvantaged communities	2022	https://nyassembly.gov/leg/?bn=S07788&term=2021
OR	Volkswagen (VW) Settlement Allocation	Executive Order 17-21, 2017	The plan (PDF) includes the development and maintenance of electric vehicle charging stations, with a focus on rural and low-income communities .	Rural and low-income communities	2017	http://www.oregon.gov/admin/pages/executive-orders.aspx
VT	Utility Electric Vehicle (EV) Program Funding Requirement	Senate Bill 99, 2023	IOUs must prioritize low-income communities in their incentive programs.	Low-income communities	2023	https://legislature.vermont.gov/bill/status/2024/S.99
VT	Electric Vehicle (EV) Charging Station Multi-Unit Dwelling (MUD) Pilot Program	Act 55, 2021	The Vermont Agency of Transportation must establish and administer, through a memorandum of understanding with the Department of Housing and Community Development (DHCD), a pilot program to support the installation of EV charging station at MUDs and affordable housing units .	Affordable housing units		https://legislature.vermont.gov/
VA	Mid-Atlantic Region Electric Vehicle (EV) Support	MAEP Website	Participating States commit to creating a regional network of EVs and EV charging stations that will make it possible to seamlessly operate light-, medium-, and heavy-duty EVs across transportation corridors and in low-income communities .	Low-income	Sep-20	https://vacleancities.org/mid-atlantic-electrification-partnership/
VA	Transportation Electrification Study	House Bill 2282, 2021	Utility and public investments that complement private efforts to deploy electric vehicle (EV) charging stations, focusing on low-income, minority, and rural communities .	Low-income, minority, and rural communities	2021	https://lis.virginia.gov/
WV	Mid-Atlantic Region Electric Vehicle (EV) Support	MAEP Website	Participating States commit to creating a regional network of EVs and EV charging stations that will make it possible to seamlessly operate light-, medium-, and heavy-duty EVs across transportation corridors and in low-income communities .	Low-income	Sep-20	https://vacleancities.org/mid-atlantic-electrification-partnership/
WI	Regional Electric Vehicle (REV) Midwest Plan	REV Midwest Partnership Announcement	Identify historically disadvantaged communities for equitable EV charging station development and EV adoption.	Disadvantaged communities		https://www.michigan.gov/whitmer/0,9309,7-387-90499_90640-569470--00.html

State incentives

Under state incentives, a diverse array of instruments and tools are employed to promote environmental awareness encompassing tax credits, financial incentives, rebates, prepaid charge cards, memberships for publicly-owned stations, educational programs, enhancements to the electric distribution system infrastructure, and numerous other strategies. These kinds of programs can be divided into 5 major categories - Tax Credit, Rebate, Public EV Charging Stations, Funds, and Educational Programs.

Sustainable Building Tax Credit (SBTC) (Energy, Minerals and Natural Resources Department, 2021) in New Mexico offers tax credit for installation of energy-efficient infrastructure to low-income residents. Programs offering rebates to incentivize the purchase and installation of Level 2/direct current fast charging (DCFC) stations or utility-owned charging stations in multi-family housing, and disadvantaged or low-income communities, include the EV Make-Ready Program (Joint Utilities of New York, 2020) in New York, Commercial Electric Vehicle Charging Station Rebate by United Illuminating (Avangrid, 2022)(UI) in Connecticut, Residential EV Charger Rebate (Burbank Water & Power, 2019) in Burbank, Plugging In Oregon Community Charging Rebates (Forth, 2023) in Oregon, Electric Vehicle Charging Pilot Program (“Electric Vehicle (EV) Charging Pilot Program,” n.d.) in Maryland Green New Deal (“L.A.’s Green New Deal | Sustainability pLAN 2019,”) and Charge Up LA! Residential (Los Angeles Department of Water and Power, 2013) in Los Angeles, Residential EV Charging Incentive Program (South Coast Air Quality Management District, 2023) in Diamond Bar, and a range of programs such as Golden State Priority Project (Center for Sustainable Energy, California Energy Commission, n.d.), Small Site Rebate (Southern California Edison, n.d.), New Construction Rebate (Southern California Edison, n.d.), Charging Infrastructure and Rebate (Southern California Edison, n.d.), and Empower EV Program (Pacific Gas and Electric Company, n.d.) in California. Clean Cars 4 All (California Air Resources Board, 2015) is another rebate program in California that also provides pre-paid charge cards. County including Cook County, and major states and cities like Maryland, New York, Dallas, and Seattle, have implemented a range of programs like Electric Vehicle Charging Stations Program (Cook County Government, 2023), Electric Vehicle Charging Pilot Program (“Electric Vehicle (EV) Charging Pilot Program,” n.d.), PlaNYC: Getting Sustainability Done (Mayor’s Office of Climate & Environmental Justice, 2022), Comprehensive Environmental and Climate Action Plan (Dallas Climate Action, City of Dallas, 2020), and Seattle City Light (City of Seattle, 1910) share the common objective of installing public EV charging stations in under-resourced and vulnerable community areas. Programs like Electric Vehicle Supply Equipment (EVSE) Grant Program (Agency of Commerce and Community Development, 2014) in Vermont, Turn-Key Installation (Southern California Edison, n.d.) in California, and It Pay\$ to Plug In (Department of Environmental Protection, 2023) and Electric Vehicle and EV Charging Station Grant Program (“Electric Vehicle and EV Charging Station Grant Program,” 2023) in New Jersey provide grants to design and install EV charging equipment. Furthermore, they fund programs to modify existing EV charging station grant programs. The state of Oregon offers a program - EV Charging Infrastructure Education (Forth, n.d.) which provides a self-guided course covering essential topics required for maximizing federal charging investments and establishing effective charging programs in underserved and rural communities. Another campaign called Electric For All (Veloz, 2018) in California provides information on charging station incentives, beneficial time-of-use rates for EV charging, and reduced vehicle registration fees and high occupancy lane exemptions

by offering a zip-code based “incentives and perks” search tool with income-based eligibility requirements.

9. Reviewer 1 comment

- Line 305, typo, “SDF units” instead of “SFD”.

Response: Thank you, we have corrected the typo in the manuscript:

“Low-income and minority people are more likely to live in MUD housing units and less likely to install a private EV charger compared to high-income groups and White people, who are more likely to live in SFD units.”

Added References in Response to R1:

- Agency of Commerce and Community Development, 2014. Electric Vehicle Supply Equipment (EVSE) Grant Program [WWW Document]. URL <https://accd.vermont.gov/community-development/funding-incentives/electric-vehicle-supply-equipment-evse-grant-program> (accessed 10.30.23).
- Avangrid, 2022. Commercial Electric Vehicle Charging Station Rebate by United Illuminating (UI) [WWW Document]. uinet. URL https://www.uinet.com/w/united-illuminating-calls-for-contractors-for-statewide-electric-vehicle-charging-program?p_1_back_url=%2Fsearch%3Fq%3DCommercial%2520Electric%2520Vehicle%2520%28EV%29%2520Charging%2520Station%2520Rebate (accessed 10.30.23).
- Barbose, G.L., Forrester, S., Darghouth, N.R., Hoen, B., 2020. Income Trends among US Residential Rooftop Solar Adopters. Lawrence Berkeley National Lab.(LBNL), Berkeley, CA (United States).
- Battelle, 2000. Travel patterns of People of Color: (440822008-001). <https://doi.org/10.1037/e440822008-001>
- Borenstein, S., Davis, L.W., 2016. The Distributional Effects of US Clean Energy Tax Credits. *Tax Policy Econ.* 30, 191–234. <https://doi.org/10.1086/685597>
- Brown, D.P., 2022. Socioeconomic and demographic disparities in residential battery storage adoption: Evidence from California. *Energy Policy* 164, 112877. <https://doi.org/10.1016/j.enpol.2022.112877>
- Burbank Water & Power, 2019. Residential EV Charging Stations Rebate [WWW Document]. Burbank Water Power. URL <https://www.burbankwaterandpower.com/electric-vehicles/residential-ev-charger-rebate> (accessed 10.30.23).
- California Air Resources Board, 2015. Clean Cars 4 All [WWW Document]. URL <https://ww2.arb.ca.gov/our-work/programs/clean-cars-4-all> (accessed 10.30.23).
- Center for Sustainable Energy, California Energy Commission, n.d. Golden State Priority Project [WWW Document]. URL <https://calevip.org/incentive-project/gssp-incentive-east-central> (accessed 10.30.23).
- City of Seattle, 1910. Seattle City Light [WWW Document]. URL <https://www.seattle.gov/city-light> (accessed 11.1.23).
- Cook County Government, 2023. Electric Vehicle Charging Stations Program [WWW Document]. Cook Cty. URL <https://www.cookcountyil.gov/EVcharging> (accessed 10.30.23).

Dallas Climate Action, City of Dallas, 2020. Comprehensive Environmental and Climate Action Plan [WWW Document]. Dallas Clim. URL <https://www.dallasclimateaction.com> (accessed 10.30.23).

Davis, L.W., 2019. Evidence of a homeowner-renter gap for electric vehicles. *Appl. Econ. Lett.* 26, 927–932. <https://doi.org/10.1080/13504851.2018.1523611>

Department of Environmental Protection, 2023. It Pay\$ to Plug In [WWW Document]. Drive Green NJ. URL <https://dep.nj.gov/drivegreen/it-pays-to-plug-in/> (accessed 10.30.23).

Electric Vehicle and EV Charging Station Grant Program, 2023. . Drive Green NJ. URL <https://dep.nj.gov/drivegreen/emobility-eligible-projects/> (accessed 10.30.23).

Electric Vehicle (EV) Charging Pilot Program [WWW Document], n.d. . Md. Public Serv. Comm. URL <https://www.psc.state.md.us/wp-content/uploads/PC44-EV-Pilot-Fact-Sheet-4.pdf> (accessed 8.30.23).

Energy, Minerals and Natural Resources Department, 2021. Sustainable Building Tax Credit (SBTC). *Energy Conserv. Manag.* URL <https://www.emnrd.nm.gov/ecmd/tax-incentives/sustainable-building-tax-credit-sbtc/> (accessed 10.30.23).

Federal Highway Administration, 2023. National Electric Vehicle Infrastructure Standards and Requirements [WWW Document]. *Fed. Regist.* URL <https://www.federalregister.gov/documents/2023/02/28/2023-03500/national-electric-vehicle-infrastructure-standards-and-requirements> (accessed 10.25.23).

FHA, 2023. National Electric Vehicle Infrastructure Formula Program Guidance (Update). U.S. Department of Transportation.

Forth, 2023. Plugging In Oregon Community Charging Rebates [WWW Document]. URL <https://forthmobility.org/Community-Charging-Rebate> (accessed 10.30.23).

Forth, n.d. Federal Charging Support [WWW Document]. URL <https://forthmobility.org/Federal-Charging-Support> (accessed 10.30.23).

Gao, X., Zhou, S., 2022. Solar adoption inequality in the U.S.: Trend, magnitude, and solar justice policies. *Energy Policy* 169, 113163. <https://doi.org/10.1016/j.enpol.2022.113163>

Geurs, K.T., van Wee, B., 2004. Accessibility evaluation of land-use and transport strategies: review and research directions. *J. Transp. Geogr.* 14.

Geverdt, D., 2019. Education Demographic and Geographic Estimates Program (EDGE): Locale Boundaries File Documentation, 2017 (No. NCES 2018-115). National Center for Education Statistics, U.S. Department of Education, Washington, DC.

Handy, S., Niemeier, D., 1997. Measuring Accessibility: An Exploration of Issues and Alternatives. *Environ. Plan. Econ. Space* 29, 1175–1194.

Joint Utilities of New York, 2020. EV Make-Ready Program [WWW Document]. URL <https://jointutilitiesofny.org/ev/make-ready> (accessed 10.30.23).

L.A.'s Green New Deal | Sustainability pLAn 2019 [WWW Document], n.d. . pLAn. URL <https://plan.lamayor.org/node> (accessed 10.30.23).

Levine, J., Grengs, J., Merlin, L.A., 2019. From Mobility to Accessibility, From Mobility to Accessibility. Cornell University Press.

Los Angeles Department of Water and Power, 2013. Charge Up LA! Residential [WWW Document]. URL https://www.ladwp.com/ladwp/faces/wcnav_externalId/r-sm-rp-ev?_afWindowId=null&_afLoop=1128087689503446&_afWindowMode=0&_adf.ctrl-state=ux4vaz2u9_4#%40%3F_afWindowId%3Dnull%26_afLoop%3D1128087689503446%26_afWindowMode%3D0%26_adf.ctrl-state%3Dhch3nr8ln_4 (accessed 11.1.23).

- Lyons, T., n.d. Incorporating Equity and Justice⁴⁰ in NEVI and Beyond.
- Maness, S.B., Merrell, L., Thompson, E.L., Griner, S.B., Kline, N., Wheldon, C., 2021. Social Determinants of Health and Health Disparities: COVID-19 Exposures and Mortality Among African American People in the United States. *Public Health Rep.* 136, 18–22. <https://doi.org/10.1177/0033354920969169>
- Mayor’s Office of Climate & Environmental Justice, 2022. PlaNYC: Getting Sustainability Done [WWW Document]. NYC Mayors Off. Clim. Environ. Justice. URL <https://climate.cityofnewyork.us/initiatives/planyc-getting-sustainability-done/> (accessed 10.30.23).
- National Electric Vehicle Infrastructure Standards and Requirements [WWW Document], 2023. . Fed. Regist. URL <https://www.federalregister.gov/documents/2023/02/28/2023-03500/national-electric-vehicle-infrastructure-standards-and-requirements> (accessed 10.6.23).
- Oliver, M., Shapiro, T., 2013. *Black Wealth / White Wealth: A New Perspective on Racial Inequality*. Routledge.
- Pacific Gas and Electric Company, n.d. Empower EV Program [WWW Document]. URL https://www.pge.com/en_US/residential/solar-and-vehicles/options/clean-vehicles/electric/empower-ev-program.page (accessed 10.30.23).
- Patterson, R., Boyd, L., Khatib, M., Taylor, T., Balik, J., Marpillero-Colomina, A., Francis, D.S., 2023. How States Can Lead on Racial and Economic Equity through the National Electric Vehicle Infrastructure (NEVI) Program.
- Smart Cities Dive, 2022. EV charging station rollout hampered by outdated state, city regulations: report [WWW Document]. Smart Cities Dive. URL <https://www.smartcitiesdive.com/news/ev-charging-stations-hampered-outdated-state-city-policies-regulations-permits/634819/> (accessed 10.7.23).
- South Coast Air Quality Management District, 2023. Residential EV Charging Incentive Program [WWW Document]. URL <https://www.aqmd.gov/home/programs/community/community-detail?title=ev-charging-incentive> (accessed 11.1.23).
- Southern California Edison, n.d. Small Site Rebate [WWW Document]. URL <https://www.sce.com/evbusiness/chargeready/small-site-rebate> (accessed 10.30.23a).
- Southern California Edison, n.d. New Construction Rebate Program [WWW Document]. URL <https://www.sce.com/evbusiness/chargeready/new-construction-rebate> (accessed 10.30.23b).
- Southern California Edison, n.d. Charging Infrastructure and Rebate Program [WWW Document]. URL <https://www.sce.com/evbusiness/chargeready/charging-infra-rebate> (accessed 10.30.23c).
- Southern California Edison, n.d. Turn-Key Installation Program [WWW Document]. URL <https://www.sce.com/evbusiness/chargeready/turn-key-installation> (accessed 10.30.23d).
- Sunter, D.A., Castellanos, S., Kammen, D.M., 2019. Disparities in rooftop photovoltaics deployment in the United States by race and ethnicity. *Nat. Sustain.* 2, 71–76. <https://doi.org/10.1038/s41893-018-0204-z>
- US Department of Transportation, 2023. Charger Types and Speeds | US Department of Transportation [WWW Document]. URL <https://www.transportation.gov/rural/ev/toolkit/ev-basics/charging-speeds> (accessed 11.7.23).

Veloz, 2018. Electric For All [WWW Document]. Electr. All. URL
<https://www.electricforall.org/> (accessed 10.30.23).

Yang, Y., Diez-Roux, A.V., 2012. Walking Distance by Trip Purpose and Population Subgroups.
Am. J. Prev. Med. 43, 11–19. <https://doi.org/10.1016/j.amepre.2012.03.015>

Response to reviewer #2

Reviewer #2 (Remarks to the Author):

1. Reviewer 2 comment

Dear Authors,

Authors should consider additional analyses to support the claims made in regressions analysis at the county level to analyze the factors contributing to the racial and income gaps in EV infrastructure accessibility. The paper should clearly outline the research objectives and hypotheses. Need to provide more details on uncertain variables used to assess income and racial disparities in EV infrastructure accessibility. The paper relies on 2021 micro-level data from a large number of households, which is commendable. However, authors should mention the data normalization. A more comprehensive research gap is needed to provide a better context for the proposed study. This suggests that local governments should implement tailored solutions, it would be helpful to provide more specific government policy recommendations based on the existing policies. How can government or policymakers address the disparities identified in the study? Overall, the paper's contribution and organization need improvements.

Response: Thank the reviewers for the thoughtful comments. As the comments are presented as a consolidated paragraph, we have taken the approach of breaking down the entire paragraph into six individual sub-comments. Our aim is to address each sub-comment in a comprehensive and detailed manner, ensuring that we adequately responded to all the points raised by the reviewer.

2. Reviewer 2 comment

Sentence 1: Authors should consider additional analyses to support the claims made in regressions analysis at the county level to analyze the factors contributing to the racial and income gaps in EV infrastructure accessibility.

Response: We appreciate the reviewer's comment. Although why these factors arise is beyond the scope of our paper, we have comprehensively investigated the potential factors contributing to racial and income disparities in EV infrastructure accessibility. We compiled a list of potential variables that could influence EV infrastructure accessibility based on an exhaustive systematic literature review. We have provided the analysis in the Supplementary Note 4. Previous studies have suggested that factors such as income, population density, proximity to highways, rates of multi-unit dwellings, and education have a significant impact on the accessibility of EV infrastructure and equity (Hsu and Fingerma, 2021a; Khan et al., 2022). This exercise in assembling this literature review strongly supports our selection of variables in the regression analysis. We also conducted an additional analysis using census data to match our county level analysis.

We integrated the results of this new robustness analysis into our manuscript in response to the reviewer's feedback. Third, we have taken the reviewer's feedback to heart and meticulously revised the manuscript. In doing so, we have adjusted our language to accurately reflect the level of support our analysis can provide, ensuring that our conclusions and claims align closely with the empirical findings. Overall, in an endeavor to maintain rigor, we have exercised caution in our articulation to avoid overstatement. The aim of our analysis is to provide an exploratory examination to discern which factor exhibits the strongest correlation and is a potential contributor

to the accessibility gap, grounded in a comprehensive systematic literature review. For further details, we invite the reviewer to review the revised manuscript and the supplementary note 4 provided below:

Manuscript (lines 331-341):

“Potential contributors to the social disparities in public EV infrastructure

Our previous analysis revealed that accessibility gaps vary widely across different counties. To shed light into the causes of these heterogeneous accessibility gaps, we delve into the potential factors that have strong correlation with social disparities in EV infrastructure accessibility, employing a regression-based framework in this section. Our approach involves examining the strong correlation between county-level accessibility gap and a set of predictor variables at the county level. Factors with strong correlation help to explain the accessibility gap. We conduct two separate OLS regressions, with one focusing on the accessibility gap in income and the other on the accessibility gap in ethnic groups. Our variables derive from a systematic literature review (see Supplementary Note 4) and our methodology can be seen in full detail in the Methods section.”

Supplementary Note 4. A systematic literature review for EV accessibility and its contributing factors (SI pages 47-49):

Using the search protocol presented in Petticrew and Roberts 2008 (Petticrew and Roberts, 2008), we conducted a comprehensive literature review, including both peer-reviewed journals and conference papers.

The search protocol was implemented using Web of Science to identify potential papers and designed capture the maximum number of papers related to electric vehicle infrastructure accessibility, with a specific focus on equity. The search terms used were (EV infrastructure OR EV charger) AND (disparity OR disparities OR inequitable OR equity OR equitable). Systematic literature reviews necessitate the explicit application of criteria to filter and include specific references. Here, we include papers if they were academic peer-reviewed, conference papers, or dissertations. Papers were excluded if they were published more than five years ago or focused on private EV infrastructure.

We identified 17 references for this review. In Supplementary Table S11, we provide an overview of how we arrived at the final set of papers and the distribution of references within our review sample.

Supplementary Table S11. Total Number of Papers Included in This Literature Review.

Key words	Outcome
EV infrastructure AND disparit*	23
EV infrastructure AND equit*	37
EV infrastructure AND inequitable	3
EV charger AND disparit*	6
EV charger AND equit*	4
EV charger AND inequitable	0
EV charging infrastructure AND disparit*	19

EV charging infrastructure AND equit*	12
EV charging infrastructure AND inequitable	3
Total	107
Duplicate	46
Unrelated and not specifically mentioned factors	41
Private EV infrastructure	3
Final number	17

We evaluated the 17 papers in our sample to assess the factors that will impact the accessibility of EV infrastructure. These factors are illustrated in Supplementary Table S12.

Supplementary Table S12. A Full List of the 17 Papers.

Papers	Methods	Level of Study	Coverage	Factors
(Roy and Law, 2022)	Machine learning	census block group	Orange County, California	Vehicles Available Population Density Poverty Education Income Household size Housing affordability Average commute time Age between 25 to 45 (%) Employment Rate PM 2.5 level Traffic impact
(Hsu and Fingerman, 2021b)	Generalized additive model	census block group	California	Distance to nearest highway or freeway Multi-unit housing unit rate Median household income
(Li et al., 2022)	Spatial autocorrelation	city level (top 10 cities)	China	GDP Population Area
(Cheng and Kontou, 2023)	A hierarchical agglomerative clustering method	trip diary data	USA	Income
(Carlton and Sultana, 2022)	Literature Review	Articles	All	Age Income Mobility
(Khan et al., 2022)	Correlation analyses	Zipcode	New York City	Median household income Race (%) Presence of highways Poverty rate
Nazari-Heris et al., 2022)	A self-scheduling model		Los Angeles	Income
(Min and Lee, 2020)		census tracts	Seattle, Washington	Income Homeowner rate
(Badiei and Do Prado, 2023)	Descriptive		USA	Income Population density
(Kempton et al., 2023)	Logit	household	Ireland	Income
(Loni and Asadi, 2023)	Equitable placement model	Chargers	San Francisco	Race Income Culture Education
(Rabinowitz et al., 2023)	Optimization model	vehicles	USA	Rural/Urban Income

(Asekomeh et al., 2021)	a positive review and value argumentation method		Dundee City, UK	Income Employment Crime and health deprivation Disability.
(Min et al., 2023)	Spatial analysis	permit, tract-level data	Seattle, WA	Gini Poverty HomeValue Income White Education SingleFamily HomeOwn Population density
(Tsukiji et al., 2023)	EV-user-centric Deployment Decision-making Model	census tracts	Los Angeles County	Gini
(Dall-Orsoletta et al., 2022)	Descriptive	Descriptive	All	Rural/Urban Income
(Desai et al., 2022)	Descriptive	county level	USA	Rural/Urban

The table offers valuable insights into the key factors discussed within the selected references. Notably, “income” emerges as the most frequently mentioned factor, indicating that researchers often emphasize the role of income levels in discussions related to electric vehicle infrastructure accessibility. This underscores the significance of economic considerations in assessing and addressing disparities in this context. Following closely behind income, “population density” stands out as another prominent factor. It suggests that the population density of an area is a critical determinant of EV infrastructure accessibility. The “rural/urban” divide also appears as a notable consideration, highlighting that the urban-rural distinction is relevant in discussions regarding EV infrastructure. “distance to nearest highway or freeway” is another factor of significance. This factor emphasizes the role of transportation infrastructure in ensuring convenient access to EV charging. The “multi-unit housing unit rate” factor suggests that the type of housing people reside in can influence their ability to access EV charging. Education and other factors also make appearances, indicating the multidimensional nature of the issue.

In summary, the table reveals a spectrum of factors that researchers have identified as relevant in the discussion of EV infrastructure accessibility and equity. These factors encompass economic, demographic, geographical, and housing-related considerations, underlining the complexity of the issue and the need for a holistic approach to achieving equity in electric vehicle infrastructure.

Building upon the insights gained from the literature review, we run regression analyzes to empirically test the influences of these key factors on the EV infrastructure accessibility gap. Our regression results show the significance of the highway gap as a primary contributor to the accessibility gap, closely followed by the income gap. Furthermore, our observations indicate that MUD (Multi-Unit Dwelling) gaps also wield a crucial influence in contributing to the accessibility gap between rural and urban settings. This can be attributed to the elevated presence of charging stations around MUDs in urban areas.

3. Reviewer 2 comment

Sentence 2: The paper should clearly outline the research objectives and hypotheses.

Response: We have added a subsection called “Research objectives and contributions” in the Methods section (lines 582-608). To the best of our knowledge this research is the first of its kind and we are establishing foundational material. The reviewer can also read the revision below:

“Research objectives and contributions

In this paper, we outline two primary research objectives. First, we are interested in quantifying the differential accessibility to public EV charging infrastructure by income and race, two critical factors that arise in the literature. Second, we aim to provide an analysis of the geographic heterogeneity of accessibility gaps across different counties and states. Finally, we seek to understand the origins of disparities and explore potential strategies for narrowing the gap. To address our first research objective, we have formulated two sub-tasks. First, we inquire whether the current EV infrastructure network offers equitable access to all residents. Second, we investigate the respective roles of income and racial disparities in shaping the accessibility of EV infrastructure. It’s noteworthy that, for our research questions, we refrain from positing specific hypotheses. We adopt an exploratory stance, presenting results as they unfold, consistent with the emerging research on EV adoption in general.

The existing body of literature consistently highlights a troubling trend: diminishing accessibility to EV infrastructure among low-income households, individuals from diverse racial backgrounds, including the Hispanic population (Bauer et al., 2021; Goetz et al., 2021; Jackson, 2021; Keith et al., 2021; Khan et al., 2022; Lee et al., 2020; Sunter et al., 2019), as well as those residing in multi-family dwellings (Hardman et al., 2021; Hsu and Fingerman, 2021a). We also know that these trends in certain locations (e.g., California and New York City) and for certain higher-level geographies (e.g., some zipcodes). There is, however, a significant gap in first, understanding trends using a highly resolved unit of analysis (i.e., individual or microscale level) and second, an understanding of how these patterns vary nationally. Our paper directly contributes to the existing body of knowledge by presenting a national analysis enriched with individual household-level data.”

4. Reviewer 2 comment

Sentence 3: Need to provide more details on uncertain variables used to assess income and racial disparities in EV infrastructure accessibility.

Response: Thank you. We are somewhat uncertain as to what the reviewer is asking for here. First and foremost, it’s essential to underscore that our dataset operates at the individual household level, which provides us detailed estimated income information as well as racial demographics. We do not need nor use additional variables to derive these measures.

In cases where households comprise multiple racial groups, consistent with research norms, we selected the head of the household’s primary racial group for classification. We identified low- and moderate-income (LMI) households by employing county-level income thresholds. These thresholds have been sourced from the Tool for Determining Low and Moderate Income (LMI) Households, a resource provided by the U.S. Department of the Treasury (U.S. Department of Treasury, 2021). We adopt a standard household size of three members, as recommended by the Tool. Lastly, our assessment of income disparities, racial disparities, and electric vehicle (EV) infrastructure accessibility is carried out at the individual household level. We subsequently

aggregate these data to county and state levels based on the granularity of our dataset, to elucidate the disparities in accessibility to EV infrastructure across varying income and racial groups. It's noteworthy that our approach is, to the best of our knowledge, the first to evaluate EV accessibility at the individual household level, underscoring the novel nature of our analysis in the existing literature.

To make these considerations clear, we revised our data section in manuscript (lines 507-516) and (lines 542-546) as follows:

“We obtained individual-level demographics data from Data Axle. To prevent double counting issues, we include only households indicating their current addresses as their primary addresses. This resulted in 120,962,661 observations, accounting for approximately 97.5% of households in the United States. We then focused our study on Black and White households, resulting in 94,878,711 observations, which accounts for 77% of the households in the United States. These data include information on the household's projected income, race, and type of dwelling. A detailed summary of the data can be found in Supplementary Table S2. This level of granularity empowers us to distinguish and categorize households into income and racial groups. In cases where households comprise multiple racial groups, we have selected the head of the household's primary racial group for classification.

2) LMI. Low- and moderate-income thresholds for each county are obtained from the Tool for Determining Low and Moderate Income (LMI) Households provided by the U.S. Department of the Treasury (U.S. Department of Treasury, 2021). To streamline our analysis, we have adopted the moderate-income threshold for a typical three-member household for each county as recommended by the Tool.”

5. Reviewer 2 comment

Sentence 4: The paper relies on 2021 micro-level data from a large number of households, which is commendable. However, authors should mention the data normalization.

Response: We are confused by this comment; we do not normalize any data. Our primary source of household-level data is derived from Data Axle, encompassing a comprehensive database of 121 million households in the United States. Notably, Data Axle has undergone a meticulous reconstruction process, integrating the most recent income statistics from the U.S. Census Bureau, self-reported survey data, real-time shifts in earned income dynamics, and updates within the Consumer Database itself. The modeling framework is grounded in a robust foundation of statistical analysis, culminating in final estimates that are adjusted at both the county and block group levels. These adjustments are made to ensure a more accurate representation of income distributions, aligning closely with the most current Census data.

Furthermore, we have taken steps to corroborate the reliability of Data Axle data, as detailed in our data section. Specifically, we conducted a correlation analysis of income and race by comparing our dataset with the US Census American Community Survey data at the zip code level. The results, as presented in Supplementary Table S4, demonstrate a high and positive correlation between our data and the Census data, affirming the quality and accuracy of our dataset.

To make this clear, we have revised the data section within our manuscript (lines 507-525), specifically in the Methods section. We also reattached the Supplementary Table S4 in this letter. The reviewer is encouraged to review the updated manuscript and the tables in the original SI document, accessible below:

“Individual data (Data Axle). We obtained individual-level demographics data from Data Axle. To prevent double counting issues, we include only households indicating their current addresses as their primary addresses. This resulted in 120,962,661 observations, accounting for approximately 97.5% of households in the United States. In section Social Disparity in County Accessibility Gaps in Public EV Infrastructure, we then focused our study on Black and White households, resulting in 94,878,711 observations, which accounts for 77% of the households in the United States. These data include information on the household’s estimated income, race, and type of dwelling. A detailed summary of the data can be found in Supplementary Table S2. This level of granularity empowers us to distinguish and categorize households into income and racial groups. In cases where households comprise multiple racial groups, we have selected the head of the household’s primary racial group for classification. Notably, Data Axle has undergone a meticulous reconstruction process, integrating the most recent income statistics from the U.S. Census Bureau, self-reported survey data, real-time shifts in earned income dynamics, and updates within the Consumer Database itself. The modeling framework is grounded in a robust foundation of statistical analysis, culminating in final estimates that are adjusted at both the county and block group levels. These adjustments are made to ensure a more accurate representation of income distributions, aligning closely with the most current Census data. To verify the reliability of Data Axle data, we conducted income and race correlation analysis with the US Census American Community Survey data at the zip code level, and found that they highly and positively correlated in Supplementary Table S4.”

Supplementary Table S4. Correlation Between Data Axle and US Census

State	White Corr	Black Corr	Asian Corr	Hispanic Corr	Median income corr
AK	0.95921055	0.85417152	0.94341309	0.933331447	0.650338893
AL	0.97226049	0.83857593	0.92548254	0.90788594	0.739773581
AR	0.97935182	0.83325368	0.79906114	0.986814867	0.503363626
AZ	0.98244511	0.89289398	0.90215464	0.977313712	0.866999318
CA	0.91107358	0.81698053	0.96376628	0.938220878	0.850046622
CT	0.97312768	0.82870866	0.91826388	0.982643448	0.905292049
CO	0.98817383	0.79480691	0.95532187	0.985198088	0.788291031
DC	0.83035865	0.92837955	0.95216129	0.96497744	0.937075619
DE	0.96411262	0.74476667	0.96257101	0.97142568	0.846878665
FL	0.93878324	0.83692158	0.89622577	0.980598313	0.891604302
GA	0.95066254	0.85235879	0.96965762	0.964424956	0.806079114
HI	0.94449117	0.7333961	0.97171689	0.84862715	0.773746233
IA	0.996035	0.8785852	0.93679651	0.96901931	0.63493101
ID	0.99423471	0.59778137	0.90061455	0.960839836	0.516256413
IL	0.94311378	0.95632385	0.96453579	0.975202279	0.813318408
IN	0.94263393	0.8091931	0.92359559	0.92468539	0.694513277
KS	0.99188782	0.77148381	0.92616198	0.986693721	0.767812856
KY	0.99183502	0.79289235	0.93440749	0.94819756	0.730818574
LA	0.90419869	0.87167619	0.8857016	0.864130638	0.622425158
MA	0.97791094	0.86015681	0.97633224	0.958523917	0.875005924
ME	0.99165067	0.67790671	0.81294636	0.784868957	0.634087267
MD	0.96798561	0.92526082	0.98040517	0.980578405	0.832056883
MI	0.98300834	0.94542536	0.94460346	0.968921821	0.842791593
MN	0.99370619	0.69912427	0.97239665	0.961681252	0.84640361
MO	0.98863276	0.89989212	0.94413208	0.940409632	0.758367483
MS	0.97241157	0.82451601	0.88345721	0.898870231	0.610623333
MT	0.99162515	0.74296009	0.76635661	0.952193597	0.308836093
NC	0.96399955	0.77111384	0.96547541	0.962381484	0.771445904
ND	0.99653985	0.86462807	0.90264125	0.936406816	0.371413088
NE	0.99441589	0.80062923	0.89296506	0.984540098	0.719011097
NH	0.99129439	0.79255351	0.94005989	0.954410956	0.783946118
NJ	0.97162663	0.90736914	0.98132722	0.987234409	0.842865105
NM	0.99047845	0.87381818	0.90739335	0.990325253	0.501309027
NV	0.96465227	0.84614703	0.97462043	0.97963474	0.733591632
NY	0.94431246	0.95418365	0.98082119	0.985251878	0.820877417
OH	0.98702063	0.8603889	0.94363887	0.965347255	0.782136038
OK	0.98780637	0.77026938	0.93650129	0.971335038	0.665797164
OR	0.98366543	0.76627578	0.95442581	0.978467986	0.663268808
PA	0.98710603	0.93678978	0.95529588	0.991716011	0.736423789

RI	0.97982131	0.95205724	0.90087865	0.982443047	0.72645938
SC	0.9723242	0.76022032	0.87929295	0.908331476	0.817105445
SD	0.99332905	0.648128	0.83283405	0.917605672	0.612321045
TN	0.98473539	0.87048789	0.93401259	0.96328216	0.845920198
TX	0.89863463	0.76893743	0.91570787	0.933683113	0.780055306
UT	0.97682944	0.58748616	0.91159731	0.977287805	0.686810014
VA	0.97072974	0.80078858	0.97745693	0.978646147	0.854067951
VT	0.97757034	0.6770943	0.89150354	0.820684032	0.582534237
WA	0.99072465	0.86822688	0.96906824	0.978816815	0.799902576
WI	0.94695403	0.91758439	0.92920591	0.964830693	0.839383333
WV	0.98389494	0.83456732	0.90009808	0.841460744	0.464961184
WY	0.99515953	0.84325299	0.74508887	0.953905359	0.449097655

6. Reviewer 2 comment

Sentence 5-7: A more comprehensive research gap is needed to provide a better context for the proposed study. This suggests that local governments should implement tailored solutions, it would be helpful to provide more specific government policy recommendations based on the existing policies. How can government or policymakers address the disparities identified in the study?

Response: We thank the reviewer for this excellent comment. The reviewer identifies a very important perspective: that there is a clear gap in regulations at the state level. Additionally, public EV infrastructure constitutes a critical market of national interest, and that this market should at least be regulated by public authorities to ensure access for all. As we note in our response to a similar question from Reviewer 1, we conducted additional research on how the current policy, both at the federal and subnational, ensures the equitable access to the public EV infrastructure, and what additional efforts the public authorities can do in the discussion section. For our comprehensive policy analysis, please refer to the Supplementary Note 3. The reviewer can also check the Supplementary Note 3 and revised manuscript as follows:

Manuscript (lines 433-448):

“A crucial aspect of this transition is the strategic development of EV charging stations in economically disadvantaged regions, ensuring that the benefits of electric transportation are accessible to all segments of society. We conducted an extensive analysis of EV charging regulations and incentives at the federal and state government levels (see Supplementary Note 3 for the comprehensive database). We found that there are only 26 existing regulations aiming to promote equitable access to electric vehicle charging infrastructure implemented in 16 states. In contrast, incentives are more common in states across the US. This indicated a clear gap of regulations from the state government which suggest that subnational government should implement tailored solution. Given that public EV infrastructure is of significant national interest, it is crucial for this market to be well-regulated by public authorities to ensure access for all.”

Supplementary Note 3. EV Charging Regulations and Policies (SI pages 41-46):

The current federal and some state policies related to EV public infrastructure emphasize increasing the availability of charging stations and providing incentives for their installation. However, states that have policies mostly focus on broader installation, with a few states placing equity considerations front and center. It is crucial to navigate the development of EV charging stations in disadvantaged regions, ensuring that the benefits of electric transportation are accessible to all members of society. We conducted a comprehensive analysis of EV charging regulations and incentives from the federal and state government. We found that there are only 26 existing regulations aiming to promote equitable access to electric vehicle charging infrastructure implemented in 16 states. In contrast, incentives are more common in states across the US. It is evident that the United States currently lacks well-defined regulations and rules to promote equitable access to electric vehicle charging infrastructure (Smart Cities Dive, 2022). This indicated a clear gap of regulations from the state government which suggest that subnational government should implement tailored solution.

Regulations/Rules

At the federal level, one of the most significant motivators of equity and access within EV charging development is the Justice40 program, which sets a goal of providing 40% of the benefits of relevant programs to disadvantaged communities (DACs). A major federal program under the umbrella of Justice40 is the National Electric Vehicle Infrastructure (NEVI) Formula Program, which requires states to submit comprehensive plans in order to receive crucial funding for deploying EV charging around important highways (Patterson et al., 2023). In its administration of this program, the federal government has both directly and indirectly prioritized equity concerns. First, states are required to touch on equity in their NEVI plans, although there is an absence of specific requirements in terms of implementation. The Federal Highway Administration implemented NEVI standards and requirements to ensure federally funded charging stations meet a certain level of accessibility, including charging network connectivity and publicly available price and availability information (“National Electric Vehicle Infrastructure Standards and Requirements,” 2023).

At the state level, there is significant variation in implementation of NEVI guidance as well as local EV infrastructure policy (Patterson et al., 2023). To start, states have taken an array of strategies to plan for Justice40 compliance. Most states that submitted NEVI plans use the Electric Vehicle Charging Justice40 Map created from the Department of Transportation and Department of Energy’s joint interim definition of DACs (Lyons, n.d.; Patterson et al., 2023). In addition, a number of states have integrated tools that utilize more specific data available to states that may not be consistent enough across the country to be used in the federal tool. These generally utilize multiple data sets to measure the climate and pollution burdens faced by each census tract or locality, considering factors like air quality, health risks, and socioeconomic demographics. States also do not have a consistent definition of what a “benefit” looks like to a DAC, and to some degree have used their NEVI plans to describe these benefits with varying degrees of specificity. State defined benefits have included air quality benefits most prevalently, as well as workforce participation and economic development from tourism (Patterson et al., 2023). Finally, some states conducted community outreach to formulate their NEVI plans. Successful approaches ranged from social media campaigns and targeted newspaper ads to regional workshops. However, one report states these activities “were lacking involvement from groups that represented low-income

communities, communities of color, and environmental justice communities” (Patterson et al., 2023). While most states did engage with the public, some only held private meetings; states also varied in the degree to which they disclosed the specifics of their outreach and incorporated feedback in NEVI plan formulation (Patterson et al., 2023).

Outside of NEVI, there has been little coordination at the state level to facilitate equitable EV regulations. Of the regulations that are in place, most are at best indirectly linked to equity and access. In our database, we are only able to find 16 states implemented regulations to ensure the equitable accessibility, see supplementary Table S10.

Supplementary Table S10. A List of Regulations among 16 States.

State	Policy Title	Law referred to	Policy related to equity	Targeted Communities	Date Started	Links/Documents
CA	Electric Vehicle (EV) Charging Station Assessment	California Public Resources Code 25229	In addition, the assessment must analyze the existing and future infrastructure needs across California, including in low-income communities .	Low-income		http://leginfo.ca.gov/
CA	Electric Vehicle (EV) Charging Station Uptime Reporting Standards	Assembly Bill 2061, 2022	The assessment must include considerations for equitable access to EV charging stations in low-, moderate-, and high-income communities .	Low-income	2022	https://leginfo.ca.gov/faces/home.xhtml
CA	Zero Electric Vehicle (ZEV) Office Authorization and Equity Assessment	Senate Bill 1251, 2022	1. Improve access to ZEVs, supporting infrastructure, and ZEV transportation options in low-income, disadvantaged, and underserved communities . 2. Reduce pollution from transportation in low-income, disadvantaged, and underserved communities .	low-income, disadvantaged, and underserved communities	2022	https://leginfo.ca.gov/faces/billNavClient.xhtml?bill_id=202120220SB1251
CA	Zero Emission Vehicle (ZEV) Initiative	California Health and Safety Code 44258.4	1. In consultation with the State Energy Resources Conservation and Development Commission, CARB prepared a funding plan (PDF) that includes a market and technology assessment, assessments of existing zero and near-zero emission funding programs, and programs that increase access to disadvantaged, low-income, and moderate-income communities and consumers. 2. Potential programs under the initiative include those involving innovative financing, car sharing, charging infrastructure in multi-unit dwellings located in disadvantaged communities , public transit, and agricultural vanpool programs.	Disadvantaged, low-income, moderate-income communities, and Multiunit Dwellings		http://www.oal.ca.gov/
CA	Zero Emission Vehicle (ZEV) Promotion Plan	Executive Order B-16, 2012, Executive Order B-48, 2018, and Executive Orders N-19-19, 2019	Update the 2016 ZEV Action plan, with a focus on low income and disadvantaged communities .	Low income and disadvantaged communities	2012	https://www.gov.ca.gov/category/executive-orders/
CA	Electric Vehicle (EV) Pilot Programs	Public Utilities Code 740.13-740.14	Priority must be given to locations in disadvantaged communities , as defined by the California Environmental Protection Agency.	Disadvantaged communities	Sep-20	https://vacleancities.org/mid-atlantic-electrification-partnership/
HI	Electric Vehicle (EV) Charging Station Rebate Program Authorization	Hawaii Revised Statutes 243-3.5, 269-72 and 269-73	The PUC must prioritize rebate awards for EV charging stations that are publicly available; serve fuel cell electric vehicle fleets; serve multiple tenants, employees, or customers; support tourism; or serve low- or moderate-income or environmental justice communities .	Low- or moderate-income or environmental justice communities		https://www.capitol.hawaii.gov/
IL	Electric Vehicle (EV) Charging Station Building Standards for Residential Developments	Senate Bill 40, 2023	1. New single-family homes or small multifamily dwellings that qualify as affordable housing must have one EV-capable parking space per dwelling. 2. Building permits for affordable housing single-family and multifamily developments must meet the following benchmarks to support the development of EV-capable parking spaces.	Affordable Housing	2023	https://www.ilga.gov/legislation/billstatus.asp?DocNum=0040&GAID=17&GA=103&DocTypeID=SB&LegID=142908&SessionID=112
IL	Regional Electric Vehicle (REV) Midwest Plan	REV Midwest Partnership Announcement	Identify historically disadvantaged communities for equitable EV charging station development and EV adoption.	Disadvantaged communities		https://www.michigan.gov/whitmer/0,9309,7-387-90499_90640-569470--00.html
IN	Regional Electric Vehicle (REV) Midwest Plan	REV Midwest Partnership Announcement	Identify historically disadvantaged communities for equitable EV charging station development and EV adoption,	Disadvantaged communities		https://www.michigan.gov/whitmer/0,9309,7-387-90499_90640-569470--00.html

IN	Utility Electric Vehicle (EV) Charging Station Pilot Program Authorization	Indiana Code 8-1-43	Utilities must also include plans to install EV charging stations in underserved and diverse communities .	Underserved and Diverse communities		http://www.in.gov/legislative/ic/code/
MD	Utility Electric Vehicle (EV) Charging Station Deployment Authorization and Requirements	House Bill 834, 2023, Public Service Commission Order 88997 – Case No. 9478, and Maryland Statutes Public Utilities Code 7-901 through 7-905	Beginning October 1, 2023, IOUs that participate in the PSC EV Pilot Program may install Level 2 EV charging stations at multi-unit dwellings in underserved communities .	Underserved Communities	2023	https://vacleancities.org/mid-atlantic-electrification-partnership/
MD	Mid-Atlantic Region Electric Vehicle (EV) Support	MAEP Website	Participating States commit to creating a regional network of EVs and EV charging stations that will make it possible to seamlessly operate light-, medium-, and heavy-duty EVs across transportation corridors and in low-income communities .	Low-income	Sep-20	https://mgaleg.maryland.gov/mgawebsite/Legislation/Details/hb0834?ys=2023RS
MA	Electric Vehicle (EV) Charging Infrastructure Deployment Support	Session Law Chapter 179, Section 81, 2022	Opportunities for EV charging stations in urban, suburban, rural, and low- and moderate-income areas .	Low- and moderate-income areas, and Rural	2022	https://malegislature.gov/Bills/192/H5060
MI	Regional Electric Vehicle (REV) Midwest Plan	REV Midwest Partnership Announcement	Identify historically disadvantaged communities for equitable EV charging station development and EV adoption.	Disadvantaged communities		https://www.michigan.gov/whitmer/0,9309,7-387-90499_90640-569470--,00.html
MN	Regional Electric Vehicle (REV) Midwest Plan	REV Midwest Partnership Announcement	Identify historically disadvantaged communities for equitable EV charging station development and EV adoption.	Disadvantaged communities		https://www.michigan.gov/whitmer/0,9309,7-387-90499_90640-569470--,00.html
NY	Zero Emission Vehicle (ZEV) Requirements	Senate Bill 7788, 2022	Strategies to accelerate deployment of affordable ZEV infrastructure that serves low-income and disadvantaged communities .	Low-income and disadvantaged communities	2022	https://nyassembly.gov/leg/?bn=S07788&term=2021
OR	Volkswagen (VW) Settlement Allocation	Executive Order 17-21, 2017	The plan (PDF) includes the development and maintenance of electric vehicle charging stations, with a focus on rural and low-income communities .	Rural and low-income communities	2017	http://www.oregon.gov/admin/pages/executive-orders.aspx
VT	Utility Electric Vehicle (EV) Program Funding Requirement	Senate Bill 99, 2023	IOUs must prioritize low-income communities in their incentive programs.	Low-income communities	2023	https://legislature.vermont.gov/bill/status/2024/S.99
VT	Electric Vehicle (EV) Charging Station Multi-Unit Dwelling (MUD) Pilot Program	Act 55, 2021	The Vermont Agency of Transportation must establish and administer, through a memorandum of understanding with the Department of Housing and Community Development (DHCD), a pilot program to support the installation of EV charging station at MUDs and affordable housing units .	Affordable housing units		https://legislature.vermont.gov/
VA	Mid-Atlantic Region Electric Vehicle (EV) Support	MAEP Website	Participating States commit to creating a regional network of EVs and EV charging stations that will make it possible to seamlessly operate light-, medium-, and heavy-duty EVs across transportation corridors and in low-income communities .	Low-income	Sep-20	https://vacleancities.org/mid-atlantic-electrification-partnership/
VA	Transportation Electrification Study	House Bill 2282, 2021	Utility and public investments that complement private efforts to deploy electric vehicle (EV) charging stations, focusing on low-income, minority, and rural communities .	Low-income, minority, and rural communities	2021	https://lis.virginia.gov/
WV	Mid-Atlantic Region Electric Vehicle (EV) Support	MAEP Website	Participating States commit to creating a regional network of EVs and EV charging stations that will make it possible to seamlessly operate light-, medium-, and heavy-duty EVs across transportation corridors and in low-income communities .	Low-income	Sep-20	https://vacleancities.org/mid-atlantic-electrification-partnership/
WI	Regional Electric Vehicle (REV) Midwest Plan	REV Midwest Partnership Announcement	Identify historically disadvantaged communities for equitable EV charging station development and EV adoption.	Disadvantaged communities		https://www.michigan.gov/whitmer/0,9309,7-387-90499_90640-569470--,00.html

State incentives

Under state incentives, a diverse array of instruments and tools are employed to promote environmental awareness encompassing tax credits, financial incentives, rebates, prepaid charge cards, both public and residential EV chargers, memberships for publicly-owned stations, educational programs, enhancements to the electric distribution system infrastructure, and numerous other strategies. These kinds of programs can be divided into 5 major categories - Tax Credit, Rebate, Public EV Charging Stations, Funds, and Educational Programs.

Sustainable Building Tax Credit (SBTC) (Energy, Minerals and Natural Resources Department, 2021) in New Mexico offers tax credit for installation of energy-efficient infrastructure to low-income residents. Programs offering rebates to incentivize the purchase and installation of Level 2/direct current fast charging (DCFC) stations or utility-owned charging stations in multi-family housing, and disadvantaged or low-income communities, include the EV Make-Ready Program (Joint Utilities of New York, 2020) in New York, Commercial Electric Vehicle Charging Station Rebate by United Illuminating (Avangrid, 2022)(UI) in Connecticut, Residential EV Charger Rebate (Burbank Water & Power, 2019) in Burbank, Plugging In Oregon Community Charging Rebates (Forth, 2023) in Oregon, Electric Vehicle Charging Pilot Program (“Electric Vehicle (EV) Charging Pilot Program,” n.d.) in Maryland Green New Deal (“L.A.’s Green New Deal | Sustainability pLAn 2019,”) and Charge Up LA! Residential (Los Angeles Department of Water and Power, 2013) in Los Angeles, Residential EV Charging Incentive Program (South Coast Air Quality Management District, 2023) in Diamond Bar, and a range of programs such as Golden State Priority Project (Center for Sustainable Energy, California Energy Commission, n.d.), Small Site Rebate (Southern California Edison, n.d.), New Construction Rebate (Southern California Edison, n.d.), Charging Infrastructure and Rebate (Southern California Edison, n.d.), and Empower EV Program (Pacific Gas and Electric Company, n.d.) in California. Clean Cars 4 All (California Air Resources Board, 2015) is another rebate program in California that also provides pre-paid charge cards. County including Cook County, and major states and cities like Maryland, New York, Dallas, and Seattle, have implemented a range of programs like Electric Vehicle Charging Stations Program (Cook County Government, 2023), Electric Vehicle Charging Pilot Program (“Electric Vehicle (EV) Charging Pilot Program,” n.d.), PlaNYC: Getting Sustainability Done (Mayor’s Office of Climate & Environmental Justice, 2022), Comprehensive Environmental and Climate Action Plan (Dallas Climate Action, City of Dallas, 2020), and Seattle City Light (City of Seattle, 1910) share the common objective of installing public EV charging stations in under-resourced and vulnerable community areas. Programs like Electric Vehicle Supply Equipment (EVSE) Grant Program (Agency of Commerce and Community Development, 2014) in Vermont, Turn-Key Installation (Southern California Edison, n.d.) in California, and It Pay\$ to Plug In (Department of Environmental Protection, 2023) and Electric Vehicle and EV Charging Station Grant Program (“Electric Vehicle and EV Charging Station Grant Program,” 2023) in New Jersey provide grants to design and install EV charging equipment. Furthermore, they fund programs to modify existing EV charging station grant programs. The state of Oregon offers a program - EV Charging Infrastructure Education (Forth, n.d.) which provides a self-guided course covering essential topics required for maximizing federal charging investments and establishing effective charging programs in underserved and rural communities. Another campaign called Electric For All (Veloz, 2018) in California provides information on charging station incentives, beneficial time-of-use

rates for EV charging, and reduced vehicle registration fees and high occupancy lane exemptions by offering a zip-code based “incentives and perks” search tool with income-based eligibility requirements.

7. Reviewer 2 comment

Sentence 8: Overall, the paper’s contribution and organization need improvements.

Response: This suggestion is vague, and we were unsure how to address. We have significantly reviewed and revised and enhanced organization of the paper, as follows,

1. Clarifying Research Objective and Hypotheses: We have clearly defined our research objectives and contributions.
2. Enhancing Data Documentation: We have improved the documentation of our data sources and methodology, in the Methods section, enhancing transparency and clarity.
3. Robustness Analysis: We have conducted additional robustness analyses (for example, by creating a novel index that captures users’ preferences for various types of charging stations) to further support our core findings and strengthen the validity of our conclusions in the Social Disparity in County Accessibility Gaps in Public EV Infrastructure section. (See Reviewer 1)
4. Regulation Insights: We have included insights into the regulatory aspects of the research, in the Discussion and Conclusion section, contributing to a more holistic understanding of the policy landscape.
5. Contribution: we have summarized the contributions of this papers as follows:

This study conducted a comprehensive national analysis of EV infrastructure accessibility, focusing on individual households at a granular level. In contributing to the existing literature, our research not only confirms some findings related to income and racial disparities but also unveils more dynamic outcomes from the equity dimension, adding new perspectives to the research community. Simultaneously, it introduces additional and more intricate dimensions to the understanding of racial disparities based on a national perspective. The study employs a robust framework for assessing EV infrastructure accessibility, considering factors such as distance, charging station preferences, and availability. Furthermore, it explores correlations with the Justice40 tool maps, providing a thorough examination of income and racial disparities in the context of equitable EV infrastructure accessibility.

We hope these components collectively contribute to a more structured and impactful manuscript, enhancing its overall quality and relevance.

Added References in Response to R2:

- Agency of Commerce and Community Development, 2014. Electric Vehicle Supply Equipment (EVSE) Grant Program [WWW Document]. URL <https://accd.vermont.gov/community-development/funding-incentives/electric-vehicle-supply-equipment-evse-grant-program> (accessed 10.29.23).
- Asekomeh, A., Gershon, O., Azubuike, S.I., 2021. Optimally Clocking the Low Carbon Energy Mile to Achieve the Sustainable Development Goals: Evidence from Dundee’s Electric Vehicle Strategy. *Energies* 14, 842. <https://doi.org/10.3390/en14040842>
- Avangrid, 2022. Commercial Electric Vehicle Charging Station Rebate by United Illuminating (UI) [WWW Document]. uinet. URL https://www.uinet.com/w/united-illuminating-calls-for-contractors-for-statewide-electric-vehicle-charging-program?p_1_back_url=%2Fsearch%3Fq%3DCommercial%2520Electric%2520Vehicle%2520%28EV%29%2520Charging%2520Station%2520Rebate (accessed 10.29.23).
- Badie, Y., Do Prado, J.C., 2023. Advancing Rural Electrification through Community-Based EV Charging Stations: Opportunities and Challenges, in: 2023 IEEE Rural Electric Power Conference (REPC). Presented at the 2023 IEEE Rural Electric Power Conference (REPC), IEEE, Cleveland, OH, USA, pp. 69–73. <https://doi.org/10.1109/REPC49397.2023.00020>
- Bauer, G., Hsu, C.-W., Nicholas, M., Lutsey, N., 2021. Charging up America: Assessing the growing need for U.S. charging infrastructure through 2030. International Council on Clean Transportation, Washington D.C.
- Burbank Water & Power, 2019. Residential EV Charging Stations Rebate [WWW Document]. Burbank Water Power. URL <https://www.burbankwaterandpower.com/electric-vehicles/residential-ev-charger-rebate> (accessed 10.29.23).
- California Air Resources Board, 2015. Clean Cars 4 All [WWW Document]. URL <https://ww2.arb.ca.gov/our-work/programs/clean-cars-4-all> (accessed 10.29.23).
- Carlton, G.J., Sultana, S., 2022. Electric vehicle charging station accessibility and land use clustering: A case study of the Chicago region. *J. Urban Mobil.* 2, 100019. <https://doi.org/10.1016/j.urbmob.2022.100019>
- Center for Sustainable Energy, California Energy Commission, n.d. Golden State Priority Project [WWW Document]. URL <https://calevip.org/incentive-project/gssp-incentive-east-central> (accessed 10.29.23).
- Cheng, X., Kontou, E., 2023. Estimating the electric vehicle charging demand of multi-unit dwelling residents in the United States. *Environ. Res. Infrastruct. Sustain.* 3, 025012. <https://doi.org/10.1088/2634-4505/acde06>
- City of Seattle, 1910. Seattle City Light [WWW Document]. URL <https://www.seattle.gov/city-light> (accessed 10.31.23).
- Cook County Government, 2023. Electric Vehicle Charging Stations Program [WWW Document]. Cook Cty. URL <https://www.cookcountyil.gov/EVcharging> (accessed 10.29.23).
- Dallas Climate Action, City of Dallas, 2020. Comprehensive Environmental and Climate Action Plan [WWW Document]. Dallas Clim. URL <https://www.dallasclimateaction.com> (accessed 10.29.23).
- Dall-Orsoletta, A., Ferreira, P., Gilson Dranka, G., 2022. Low-carbon technologies and just energy transition: Prospects for electric vehicles. *Energy Convers. Manag.* X 16, 100271. <https://doi.org/10.1016/j.ecmx.2022.100271>

Department of Environmental Protection, 2023. It Pay\$ to Plug In [WWW Document]. Drive Green NJ. URL <https://dep.nj.gov/drivegreen/it-pays-to-plug-in/> (accessed 10.29.23).

Desai, J., Mathew, J.K., Li, H., Bullock, D.M., 2022. Using Connected Vehicle Data for Assessing Electric Vehicle Charging Infrastructure Usage and Investment Opportunities. Electric Vehicle and EV Charging Station Grant Program, 2023. . Drive Green NJ. URL <https://dep.nj.gov/drivegreen/emobility-eligible-projects/> (accessed 10.29.23).

Electric Vehicle (EV) Charging Pilot Program [WWW Document], n.d. . Md. Public Serv. Comm. URL <https://www.psc.state.md.us/wp-content/uploads/PC44-EV-Pilot-Fact-Sheet-4.pdf> (accessed 8.30.23).

Energy, Minerals and Natural Resources Department, 2021. Sustainable Building Tax Credit (SBTC). Energy Conserv. Manag. URL <https://www.emnrd.nm.gov/ecmd/tax-incentives/sustainable-building-tax-credit-sbtc/> (accessed 10.29.23).

Federal Highway Administration, 2023. National Electric Vehicle Infrastructure Standards and Requirements [WWW Document]. Fed. Regist. URL <https://www.federalregister.gov/documents/2023/02/28/2023-03500/national-electric-vehicle-infrastructure-standards-and-requirements> (accessed 10.25.23).

FHA, 2023. National Electric Vehicle Infrastructure Formula Program Guidance (Update). U.S. Department of Transportation.

Forth, 2023. Plugging In Oregon Community Charging Rebates [WWW Document]. URL <https://forthmobility.org/Community-Charging-Rebate> (accessed 10.29.23).

Forth, n.d. Federal Charging Support [WWW Document]. URL <https://forthmobility.org/Federal-Charging-Support> (accessed 10.29.23).

Goetz, M., Levandowski, R., Bradbury, J., Grace Van Horn, 2021. Towards Equitable and Transformative Investments in EV Charging Infrastructure. Georgetown Climate Center, & M.J. Bradley & Associates.

Hardman, S., Fleming, K., Kare, E., Ramadan, M., 2021. A perspective on equity in the transition to electricvehicle. MIT Sci. Policy Rev. 46–54. <https://doi.org/10.38105/spr.e10rdoaoup>

Hsu, C.-W., Fingerman, K., 2021a. Public electric vehicle charger access disparities across race and income in California. *Transp. Policy* 100, 59–67. <https://doi.org/10.1016/j.tranpol.2020.10.003>

Hsu, C.-W., Fingerman, K., 2021b. Public electric vehicle charger access disparities across race and income in California. *Transp. Policy* 100, 59–67. <https://doi.org/10.1016/j.tranpol.2020.10.003>

Jackson, C.T., 2021. Expanding access to electric vehicles in California’s low-income communities. *J. Sci. Policy Gov.* 18. <https://doi.org/10.38126/JSPG180107>

Joint Utilities of New York, 2020. EV Make-Ready Program [WWW Document]. URL <https://jointutilitiesofny.org/ev/make-ready> (accessed 10.29.23).

Keith, D., Long, J., Gaiarin, B., 2021. Access to Electric Vehicle Charging in the United States. Mobilize.ai & Toyota Mobility Foundation.

Kempton, W., Pearre, N.S., Guensler, R., Elango, V.V., 2023. Influence of Battery Energy, Charging Power, and Charging Locations upon EVs’ Ability to Meet Trip Needs. *Energies* 16, 2104. <https://doi.org/10.3390/en16052104>

Khan, H.A.U., Price, S., Avraam, C., Dvorkin, Y., 2022. Inequitable access to EV charging infrastructure. *Electr. J.* 35, 107096. <https://doi.org/10.1016/j.tej.2022.107096>

- L.A.'s Green New Deal | Sustainability pLAn 2019 [WWW Document], n.d. . pLAn. URL <https://plan.lamayor.org/node> (accessed 10.29.23).
- Lee, J.H., Chakraborty, D., Hardman, S.J., Tal, G., 2020. Exploring electric vehicle charging patterns: Mixed usage of charging infrastructure. *Transp. Res. Part Transp. Environ.* 79, 102249. <https://doi.org/10.1016/j.trd.2020.102249>
- Li, G., Luo, T., Song, Y., 2022. Spatial equity analysis of urban public services for electric vehicle charging—Implications of Chinese cities. *Sustain. Cities Soc.* 76, 103519. <https://doi.org/10.1016/j.scs.2021.103519>
- Loni, A., Asadi, S., 2023. Data-driven equitable placement for electric vehicle charging stations: Case study San Francisco. *Energy* 282, 128796. <https://doi.org/10.1016/j.energy.2023.128796>
- Los Angeles Department of Water and Power, 2013. Charge Up LA! Residential [WWW Document]. URL https://www.ladwp.com/ladwp/faces/wcnav_externalId/r-sm-rp-ev?_afWindowId=null&_afLoop=1128087689503446&_afWindowMode=0&_adf.ctrl-state=ux4vaz2u9_4#%40%3F_afWindowId%3Dnull%26_afLoop%3D1128087689503446%26_afWindowMode%3D0%26_adf.ctrl-state%3Dhch3nr8ln_4 (accessed 10.31.23).
- Lyons, T., n.d. Incorporating Equity and Justice40 in NEVI and Beyond. Mayor's Office of Climate & Environmental Justice, 2022. PlaNYC: Getting Sustainability Done [WWW Document]. NYC Mayors Off. Clim. Environ. Justice. URL <https://climate.cityofnewyork.us/initiatives/planyc-getting-sustainability-done/> (accessed 10.29.23).
- Min, Y., Lee, H.W., 2020. Social Equity of Clean Energy Policies in Electric-Vehicle Charging Infrastructure Systems, in: *Construction Research Congress 2020*. Presented at the Construction Research Congress 2020, American Society of Civil Engineers, Tempe, Arizona, pp. 221–229. <https://doi.org/10.1061/9780784482858.025>
- Min, Y., Lee, H.W., Hurvitz, P.M., 2023. Clean energy justice: Different adoption characteristics of underserved communities in rooftop solar and electric vehicle chargers in Seattle. *Energy Res. Soc. Sci.* 96, 102931. <https://doi.org/10.1016/j.erss.2022.102931>
- National Electric Vehicle Infrastructure Standards and Requirements [WWW Document], 2023. . Fed. Regist. URL <https://www.federalregister.gov/documents/2023/02/28/2023-03500/national-electric-vehicle-infrastructure-standards-and-requirements> (accessed 10.6.23).
- Pacific Gas and Electric Company, n.d. Empower EV Program [WWW Document]. URL https://www.pge.com/en_US/residential/solar-and-vehicles/options/clean-vehicles/electric/empower-ev-program.page (accessed 10.29.23).
- Patterson, R., Boyd, L., Khatib, M., Taylor, T., Balik, J., Marpillero-Colomina, A., Francis, D.S., 2023. How States Can Lead on Racial and Economic Equity through the National Electric Vehicle Infrastructure (NEVI) Program.
- Petticrew, M., Roberts, H., 2008. *Systematic Reviews in the Social Sciences: A Practical Guide*. John Wiley & Sons.
- Rabinowitz, A.I., Smart, J.G., Coburn, T.C., Bradley, T.H., 2023. Assessment of Factors in the Reduction of BEV Operational Inconvenience. *IEEE Access* 11, 30486–30497. <https://doi.org/10.1109/ACCESS.2023.3255103>

- Roy, A., Law, M., 2022. Examining spatial disparities in electric vehicle charging station placements using machine learning. *Sustain. Cities Soc.* 83, 103978. <https://doi.org/10.1016/j.scs.2022.103978>
- Smart Cities Dive, 2022. EV charging station rollout hampered by outdated state, city regulations: report [WWW Document]. Smart Cities Dive. URL <https://www.smartcitiesdive.com/news/ev-charging-stations-hampered-outdated-state-city-policies-regulations-permits/634819/> (accessed 10.6.23).
- South Coast Air Quality Management District, 2023. Residential EV Charging Incentive Program [WWW Document]. URL <https://www.aqmd.gov/home/programs/community/community-detail?title=ev-charging-incentive> (accessed 10.31.23).
- Southern California Edison, n.d. Small Site Rebate [WWW Document]. URL <https://www.sce.com/evbusiness/chargeready/small-site-rebate> (accessed 10.29.23a).
- Southern California Edison, n.d. New Construction Rebate Program [WWW Document]. URL <https://www.sce.com/evbusiness/chargeready/new-construction-rebate> (accessed 10.29.23b).
- Southern California Edison, n.d. Charging Infrastructure and Rebate Program [WWW Document]. URL <https://www.sce.com/evbusiness/chargeready/charging-infra-rebate> (accessed 10.29.23c).
- Southern California Edison, n.d. Turn-Key Installation Program [WWW Document]. URL <https://www.sce.com/evbusiness/chargeready/turn-key-installation> (accessed 10.29.23d).
- Sunter, D.A., Castellanos, S., Kammen, D.M., 2019. Disparities in rooftop photovoltaics deployment in the United States by race and ethnicity. *Nat. Sustain.* 2, 71–76. <https://doi.org/10.1038/s41893-018-0204-z>
- Tsukiji, T., Zhang, N., Jiang, Q., He, B.Y., Ma, J., 2023. A Multifaceted Equity Metric System for Transportation Electrification. *IEEE Open J. Intell. Transp. Syst.* 4, 690–707. <https://doi.org/10.1109/OJITS.2023.3311689>
- U.S. Department of Treasury, 2021. Tool for Determining Low and Moderate Income (LMI) Households.
- Veloz, 2018. Electric For All [WWW Document]. *Electr. All.* URL <https://www.electricforall.org/> (accessed 10.29.23).

REVIEWERS' COMMENTS

Reviewer #1 (Remarks to the Author):

I am satisfied with the authors' answers to my questions and suggestions. I think this paper can be published as is.

Reviewer #2 (Remarks to the Author):

The paper has been modified and all the comments are addressed.